# Intestinal pathogens override hunger-driven decision-making via immune regulation of central serotonin signaling in *C. elegans*

Ying Lei [1,3], Chao Chen [2,3], Xu Zhan [1], Mingshu Xie[1], Ying Wang[1], Hao Li[1], Jiale Zhang[1] & Ping Liu [1] ✉

Animals integrate internal states to guide survival-critical decisions, but whether and how intestinal bacteria influence this process by interacting with host metabolic cues remains unclear. Here we show that the intestinal pathogen *P. aeruginosa* overrides host decision-making in fasted *C. elegans* by modulating central serotonin (5-HT) signaling. Fasting promotes risk-taking by activating an intestinal energy-sensing pathway that induces the chemoreceptor SRI-36 in ADF 5-HT neurons, sensitizing ADF to food odors and triggering moderate 5-HT release that drives food attraction despite risk. In contrast, intestinal *P. aeruginosa* reverses this strategy by activating a distinct immune-brain axis that further amplifies SRI-36 expression and ADF sensitivity, leading to excessive 5-HT release that suppresses food attraction and prioritizes safety. These findings reveal a gut-to-brain mechanism by which metabolic and immune signals converge on central 5-HT to reshape behavioral strategies.

To survive in dynamic environments filled with both threats and rewards, animals must detect and integrate external sensory cues to make adaptive behavioral choices[1,2]. This process, known as multisensory threat-reward decision-making, is strongly shaped by internal states, which optimize behavioral responses to meet survival needs. For example, hunger often biases decisions toward risk-taking, prioritizing food acquisition despite potential threats[3–6]. One effective way internal states influence decision-making is through sensory modulation. Feeding states can alter sensory neuron sensitivity, rapidly shifting sensory perception and behavioral preferences[7–9]. While such adaptive mechanisms promote survival, maladaptive changes in sensory processing and impaired threat-reward decision-making are hallmarks of neuropsychiatric disorders such as autism, schizophrenia, anxiety, and depression[2,10,11]. Yet, how internal states regulate sensory processing to guide decision-making remains poorly understood.

Among internal cues, intestinal bacteria are increasingly recognized as potent modulators of brain function and behavior[12,13]. Intestinal pathogens, in particular, can elicit profound neural changes, including shifts in sensory preferences and the development of anhedonia[14,15], a core symptom of neuropsychiatric disorders characterized by the inability to experience reward[16]. Dysbiosis of the gut microbiota is associated with multiple neuropsychiatric conditions involving impaired decision-making[13,17,18]. Despite these associations, the mechanisms by which intestinal bacteria influence host decision-making remain elusive. In particular, whether they interact with metabolic states to regulate sensory processing and behavioral choices is unknown.

The nematode *C. elegans* provides a powerful genetic model for dissecting host-microbe interactions and their impact on neural function and behavior[19,20]. Its simple intestinal anatomy and compact, well-defined nervous system facilitate mechanistic studies using monoxenic bacterial cultures[19,21]. *C. elegans* also exhibits multisensory decision-making behaviors[22]. For example, when facing a hyperosmotic glycerol barrier placed before the attractive food odor diacetyl, worms must decide whether to risk crossing the potentially lethal barrier to reach the reward[6,23]. This simple form of decision-

[1]Department of Pathophysiology, Key Laboratory of Ministry of Education of China and Hubei Province for Neurological Disorders, School of Basic Medicine, Tongji Medical College, Huazhong University of Science and Technology, Wuhan, Hubei, China. [2]Department of Orthopaedics, Union Hospital, Tongji Medical College, Huazhong University of Science and Technology, Wuhan, Hubei, China. [3]These authors contributed equally: Ying Lei, Chao Chen. ✉e-mail: pingl@hust.edu.cn

making is state-dependent, as hunger enhances risk-taking[6,24]. Glycerol and diacetyl are detected by the primary sensory neurons ASH and AWA, respectively[6,25], and downstream interneurons integrate these inputs to drive behavior[26]. However, whether intestinal pathogens alter these processes and how they may interact with metabolic states to regulate behavioral choices remains unknown.

Here, we show that the intestinal pathogen *P. aeruginosa* overrides hunger-driven decision-making in *C. elegans* by modulating central serotonin (5-HT) signaling through an immune-brain axis. Mechanistically, fasting promotes risk-taking by activating the intestinal energy sensor AAK-2/AMPK, which promotes DAF-16/FOXO-dependent release of the insulin-like peptide (ILP) INS-37 from intestinal epithelial cells (IECs). INS-37 then engages the canonical DAF-2-DAF-16 insulin/IGF-1 signaling (IIS) cascade in ADF 5-HT neurons. This FOXO-to-FOXO signaling induces expression of the chemoreceptor SRI-36, conferring diacetyl sensitivity to ADF, which are otherwise insensitive in the well-fed state. Diacetyl-evoked ADF activation triggers moderate 5-HT release that inhibits AIB and activates AUA interneurons via the 5-HT receptors SER-4 and LGC-50, respectively, promoting diacetyl attraction despite the associated risk. In contrast, intestinal *P. aeruginosa* subverts this hunger-driven decision-making by activating the intestinal PMK-2/p38 MAPK immune pathway, which induces a distinct ILP, INS-7, to further amplify SRI-36 expression and ADF responsiveness through the same FOXO-to-FOXO axis. This leads to excessive 5-HT release that predominantly inhibits AIA and AIY interneurons via the 5-HT receptor MOD-1, suppressing diacetyl attraction and prioritizing safety.

Together, these findings establish *P. aeruginosa* as an intestinal pathogen capable of overriding host decision-making through immune modulation of central 5-HT signaling, and identify central 5-HT as a convergence target for intestinal metabolic and immune cues. Our study provides a mechanistic framework for how intestinal microbes and internal physiological states dynamically modulate brain function and behavioral decisions.

## Results

### Intestinal pathogen *P. aeruginosa* reverses hunger-driven decision-making via ADF 5-HT neurons in *C. elegans*

To investigate how intestinal pathogens influence host decision-making, we employed a multisensory behavioral choice assay in *C. elegans*[6,23]. In this assay, worms are attracted to the food odor diacetyl but must decide whether to cross a hyperosmotic glycerol barrier to reach it (Supplementary Fig. 1a). We tested three paradigmatic human pathogenic bacteria, *Pseudomonas aeruginosa* (*P. aeruginosa*), *Staphylococcus aureus* (*S. aureus*), and enteropathogenic *E. coli* (EPEC). However, none of these pathogens altered the behavioral choice under well-fed conditions, as escape percentages remained comparable to those of control worms fed nonpathogenic *E. coli* (Supplementary Fig. 1b).

Because hunger is known to promote risk-taking[3–6], we adapted the assay by fasting worms before the choice test (Fig. 1a). Fasting increased escape probability in a time-dependent manner across all glycerol concentrations tested, with the strongest effect observed after 6 h of fasting at 3 M glycerol (Supplementary Fig. 1c), which was selected for subsequent experiments. Under these conditions, *P. aeruginosa* infection significantly reduced escape probability compared to *E. coli*, returning it to levels similar to well-fed controls, whereas *S. aureus* and EPEC had no effect (Fig. 1b). In contrast, *P. aeruginosa* did not affect unisensory decision-making, as chemotaxis toward diacetyl (in the absence of glycerol) and escape from glycerol (in the absence of diacetyl) remained unchanged (Supplementary Fig. 1d, e). Locomotor activity was also unaffected (Supplementary Fig. 1f). These results suggest that *P. aeruginosa* selectively reverses hunger-driven risky decision-making in *C. elegans*. Similar escape suppression occurred after 4 or 24 h of infection (Supplementary Fig. 1g), with 4 h chosen for subsequent experiments.

Previous studies have suggested that *P. aeruginosa* metabolites may modulate *C. elegans* behavior via chemosensory neurons[27,28]. However, exposure to *P. aeruginosa* supernatant, odors, heat-killed cells, or a nonpathogenic mutant *gacA(−)* did not affect fasting-induced escape (Fig. 1c). In contrast, infection with live cells alone was sufficient (Fig. 1c), and enhancing virulence by supplementing the medium with 0.35% peptone[29] increased *P. aeruginosa* colonization and escape suppression (Fig. 1d and Supplementary Fig. 1h). Knockdown of *nol-6*, which reduces intestinal colonization of *P. aeruginosa*[30] (Fig. 1e), attenuated its effect on escape (Fig. 1f). Consistent with the similar escape behavior observed at 4 h and 24 h of infection, intestinal *P. aeruginosa* colonization did not differ significantly between these time points (Supplementary Fig. 1g, i). These results suggest that intestinal colonization of virulent *P. aeruginosa* is required to suppress hunger-driven decision-making.

5-HT and dopamine are key neuromodulators that regulate cognitive processes, including threat and reward processing[22,31]. Loss of the dopamine biosynthetic gene *cat-2*[32] had no effect on escape in the fasted or infected state (Fig. 1g). However, loss of the 5-HT biosynthetic gene *tph-1*[32] reduced fasting-induced escape and reversed the *P. aeruginosa*-induced suppression of escape (Fig. 1h). Additionally, *mod-5(n822)* mutants, which have elevated extracellular 5-HT due to impaired reuptake[33], showed exaggerated behavioral responses to both hunger and infection (Fig. 1i). These results suggest that both hunger and *P. aeruginosa* regulate decision-making through 5-HT signaling.

To identify the relevant 5-HT neurons, we silenced *tph-1*-expressing neurons NSM, ADF, or HSN[34] using the histamine-gated chloride channel HisCl1[35]. Silencing NSM or HSN had no effect (Fig. 1k, l), but silencing ADF significantly impaired both fasting-induced escape and *P. aeruginosa*-mediated suppression (Fig. 1j). Consistently, expressing wild-type *tph-1* in ADF, but not NSM, fully rescued the effects of both fasting and *P. aeruginosa* (Fig. 1h). Together, these results suggest that hunger promotes risky decision-making, while *P. aeruginosa* overrides this decision, both via ADF 5-HT neurons.

In addition to 5-HT, ADF neurons are also cholinergic[36]. However, ADF-specific *unc-17* RNAi, which reduces vesicular acetylcholine release[37], did not significantly affect escape in either the fasted or infected state (Supplementary Fig. 1j). These results suggest that acetylcholine release from ADF is not required for the state-dependent regulation of decision-making, although it may contribute to other ADF-mediated functions.

### ADF neurons act as primary olfactory sensory neurons for diacetyl in fasted and infected states

We next investigate how hunger and *P. aeruginosa* infection influence decision-making via 5-HT signaling. ASH and AWA sensory neurons, which detect glycerol and diacetyl respectively[6,25], form direct synaptic connections with ADF[38] (Supplementary Fig. 2a). We hypothesized that changes in these upstream neurons might influence ADF responses under different internal states. Indeed, fasting reduced glycerol-evoked calcium responses in ASH and enhanced diacetyl-evoked responses in AWA (Supplementary Fig. 2b, c). However, *P. aeruginosa* infection did not further alter ASH or AWA responses in fasted worms (Supplementary Fig. 2b, c). Silencing AWA with HisCl1 reduced escape in both well-fed and fasted states, consistent with its role in sensing diacetyl, but fasted worms still exhibited substantial escape probability compared to well-fed controls (Supplementary Fig. 2d). These results suggest that while ASH and AWA sensory responses are modulated by hunger, they are not the primary sites through which *P. aeruginosa* infection overrides hunger-driven decision-making.

ADF neurons are 5-HT sensory neurons that respond to noxious stimuli in well-fed worms[39] and to food odors under dietary restriction[40]. In a transgenic strain expressing GCaMP6 in ADF, which exhibited normal escape behavior compared to wild type

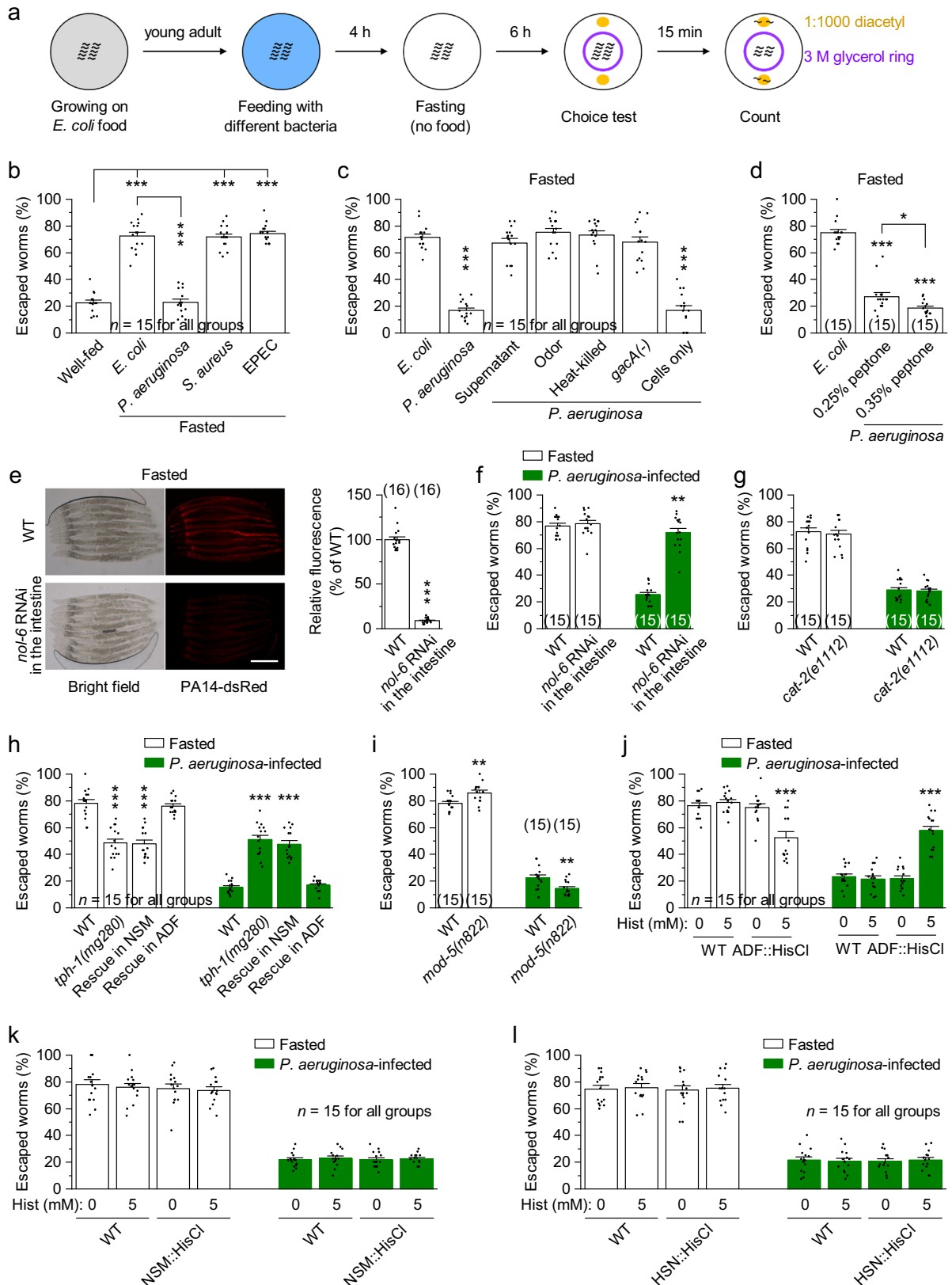

(Supplementary Fig. 2e), ADF showed no response to glycerol but were robustly activated by diacetyl (Fig. 2a, b), with responses enhanced by fasting and further amplified by *P. aeruginosa* infection (Fig. 2b). In *unc-13(e51)* mutants, which are defective in neurotransmitter release[41], diacetyl-evoked ADF responses were absent in well-fed worms but persisted in fasted and infected states, albeit at a reduced amplitude

(Fig. 2c). In contrast, *unc-31(e169)* mutants, which are defective in neuropeptide release[42,43], showed no such defect (Fig. 2c), Moreover, in *odr-7(ky4)* mutants, where AWA neurons were eliminated[44], and in *odr-7(ky4);unc-13(e51)* double mutants, ADF responses to diacetyl were abolished in the well-fed state but retained in both fasted and infected states (Fig. 2d). These results suggest that while ADF neurons receive

**Fig. 1 | Intestinal pathogen *P. aeruginosa* reverses hunger-driven decision-making via ADF 5-HT neurons in *C. elegans*. a** Schematic of the multisensory behavioral choice assay for fasted *C. elegans*. **b** Escape percentages of well-fed or fasted worms. Well-fed worms were maintained on nonpathogenic *E. coli*, and fasted worms were fed *E. coli* or pathogenic bacteria (*P. aeruginosa*, *S. aureus*, or EPEC) before fasting. $p = 0, 0, 0$, and $<0.0001$. **c** Escape percentages of fasted worms fed *E. coli*, *P. aeruginosa*, or *P. aeruginosa* components, including supernatant, odors, heat-killed cells, the nonpathogenic *gacA(−)* mutant, or live cells only. $p = 0, 0.953, 0.9809, 0.9997, 0.981$, and 0. **d** Escape percentages of fasted worms fed *E. coli* or *P. aeruginosa* with 0.35% peptone to enhance virulence. $p = 0, 0.0365$, and 0. **e** Intestinal colonization of PA14-dsRed in the indicated genotypes. Shown are representative images and comparisons of relative fluorescence normalized to wild type (WT). Intestine-specific RNAi was performed using *Pges-1*. Scale bar, 100 μm. $p < 0.0001$. **f–i** Escape percentages of fasted (fed *E. coli* before fasting) or *P.* *aeruginosa*-infected (fed *P. aeruginosa* before fasting) worms with the indicated genotypes. NSM- and ADF-specific rescue was performed using *Pceh-2* and *Psrh-142*, respectively. $p = 0.5963$ and $<0.0001$ (**f**), 0.6745 and 0.7903 (**g**), 0, 0, 0.948, $<0.0001$, $<0.0001$, and 0.9677 (**h**), and 0.0035 and 0.0036 (**i**). **j–l** Escape percentages of fasted or *P. aeruginosa*-infected worms with or without 5 mM histamine (Hist) treatment. HisCl1 was expressed in ADF (**j**), NSM (**k**), and HSN (**l**) using *Psrh-142*, *Pceh-2*, and *Pclh-3*, respectively. $p = 0.9117, 0.9914, <0.0001, 0.9691, 0.9791$, and $<0.0001$ (**j**), 0.9691, 0.9156, 0.7535, 0.9251, 1, and 0.9661 (**k**), and 0.9887, 0.9979, 0.9977, 0.9894, 0.9928, and 1 (**l**). $*p < 0.05$, $**p < 0.01$, and $***p < 0.001$ (one-way ANOVA with Tukey's post hoc test for **b–d**, **h**, **j–l**; two-sided unpaired *t*-test for **e–g**, **i**). Brackets indicate the number of independent assays (**b–d**, **f–l**) or animals tested (**e**). Data are shown as means ± SEM. Source data are provided as a Source Data file.

synaptic input from AWA in response to diacetyl stimulation, they function as primary olfactory neurons for diacetyl during fasting and *P. aeruginosa* infection, independent of synaptic input.

Olfactory detection in *C. elegans* is mediated by G-protein-coupled receptors (GPCRs) and downstream transduction channels[45]. ODR-3/Gα mediates food odor sensation in ADF neurons[40], and the TRPV channel subunits OCR-2 and OSM-9, which are coexpressed in ADF, likely function downstream of GPCR signaling[45]. In *odr-7(ky4)* mutants, mutations in *odr-3*, *ocr-2*, or *osm-9* abolished ADF responses to diacetyl in both fasted and infected worms, and ADF-specific expression of the corresponding wild-type genes rescued the defect (Fig. 2e). These results suggest that an unidentified GPCR, acting upstream of the ODR-3-OCR-2/OSM-9 signaling pathway, mediates ADF responses to diacetyl during fasting and *P. aeruginosa* infection.

## Hunger- and infection-induced SRI-36 acts as a diacetyl receptor in ADF neurons

To identify the GPCR responsible for ADF diacetyl sensitivity, we first tested ODR-10, the only known diacetyl receptor, which is exclusively expressed in AWA neurons of well-fed worms[46]. However, ADF responses to diacetyl remained comparable between *unc-13(e51)* and *unc-13(e51);odr-10(ky32)* mutants across all states (Fig. 3a), indicating that ODR-10 is not required for ADF sensitivity in the absence of synaptic input.

We next analyzed published RNA-sequencing datasets for chemosensory GPCRs upregulated during fasting or *P. aeruginosa* infection. We identified 126 candidate GPCRs induced by fasting (Fasted vs. Well-fed)[47] and 1,667 genes upregulated by *P. aeruginosa* exposure (*P. aeruginosa* vs. *E. coli*)[48], with four GPCRs, *sri-36*, *srr-6*, *str-163*, and *srap-1*, shared between comparisons (Fig. 3b). Screening these genes in *P. aeruginosa*-infected *odr-7(ky4)* mutants revealed that ADF-specific RNAi of *sri-36*, but not the other candidates, abolished ADF calcium responses to diacetyl (Fig. 3c). This effect was confirmed in fasted *odr-7(ky4)* mutants (Fig. 3d).

To further validate the role of SRI-36, we generated a CRISPR deletion allele, *sri-36(plc921)* (Supplementary Fig. 3a), which eliminated ADF responses to diacetyl in both fasted and infected *odr-7(ky4)* mutants (Fig. 3e). This defect was rescued by ADF-specific expression of wild-type *sri-36* (Fig. 3e). The deletion also impaired escape in both fasted and infected worms, which was rescued by ADF-specific expression of wild-type *sri-36* and recapitulated by ADF-specific RNAi (Fig. 3f). A *Psri-36::GFP* transcriptional reporter was undetectable in well-fed worms but strongly induced in ADF during fasting and further enhanced by *P. aeruginosa* infection (Fig. 3g and Supplementary Fig. 3b). Consistently, a CRISPR knock-in reporter, SRI-36::mNeonGreen, with mNeonGreen inserted at the C-terminus of the endogenous locus, showed a similar ADF-specific induction pattern (Supplementary Fig. 3c). To test whether receptor abundance modulates ADF sensitivity, we expressed SRI-36 at two different levels in *sri-36(plc921)* mutants by injecting 5 ng/μl (low) or 75 ng/μl (high) of *Psrh-*

*142::sri-36::mStrawberry* plasmid (Supplementary Fig. 3d). In the fasted state, low-level expression restored ADF diacetyl responses and escape to control levels (Fig. 3e, f). In contrast, high-level expression enhanced ADF diacetyl responses and suppressed escape, phenocopying the behavioral pattern observed during *P. aeruginosa* infection (Fig. 3e, f). These results suggest that SRI-36 is induced in ADF by hunger and infection and is required for diacetyl sensation, and that its expression level tunes ADF sensitivity and behavioral output.

To test whether SRI-36 functions as a diacetyl receptor, we performed multiple receptor validation assays[29,49]. First, both an SRI-36::GFP translational fusion and the CRISPR-generated endogenous SRI-36::mNeonGreen fusion protein localized to the sensory cilia of ADF (Fig. 3h and Supplementary Fig. 3c), supporting its role in detecting environmental odors. Second, ectopic expression of *sri-36* in ASH nociceptive neurons, which normally do not respond to the attractive food odor diacetyl, conferred robust diacetyl-evoked calcium responses in ASH (Supplementary Fig. 3e), demonstrating that SRI-36 is sufficient to mediate diacetyl sensitivity. Third, because ASH activation drives avoidance[25], we assessed behavioral consequences using a two-choice chemotaxis quadrant assay[29]. Ectopic *sri-36* expression in ASH significantly reduced diacetyl attraction (Supplementary Fig. 3f). Notably, in *odr-7(ky4)* mutants, ASH-expressed *sri-36* even converted diacetyl attraction to aversion (Supplementary Fig. 3f), further supporting the role of SRI-36 in mediating diacetyl sensation. Together, these results establish SRI-36 as a diacetyl receptor or essential receptor subunit in ADF neurons.

## Hunger- and infection-induced SRI-36 expression in ADF requires intestinal AAK-2/AMPK and PMK-2/p38 MAPK pathways

To investigate how fasting and *P. aeruginosa* infection regulate ADF diacetyl sensitivity, we examined the roles of the AAK-2/AMPK energy sensor and the PMK-1/p38 MAPK immune pathway. AMPK is a key metabolic sensor that responds to nutrient deprivation[40,50]. In *C. elegans*, AAK-2/AMPK phosphorylates and activates the transcription factor DAF-16/FOXO, driving DAF-16-dependent gene expression[40,51]. In fasted *odr-7(ky4)* mutants, *aak-2* deletion abolished diacetyl-evoked calcium responses in ADF, which were restored by intestinal expression of wild-type *aak-2* and recapitulated by intestine-specific RNAi of *aak-2* (Fig. 4a). Similarly, intestine-specific RNAi of *daf-16* in *odr-7(ky4)* mutants reduced ADF responses to levels comparable to *odr-7(ky4);aak-2(ok524)* mutants (Fig. 4a). Fasting-induced nuclear translocation of DAF-16 was also reduced by intestine-specific RNAi of *aak-2* (Fig. 4b), whereas intestinal overexpression of a constitutively nuclear-localized DAF-16 (DAF-16a^AM)[52] restored ADF responses in fasted *odr-7(ky4);aak-2(ok524)* mutants (Fig. 4a). The intestine-specific RNAi of *aak-2* and *daf-16* produced similar results in the systemic RNAi-defective *sid-1(qt9)* mutant background[53] (Fig. 4a, b), confirming tissue specificity. These results suggest that AAK-2 activates DAF-16 in IECs to drive ADF diacetyl sensitivity during fasting.

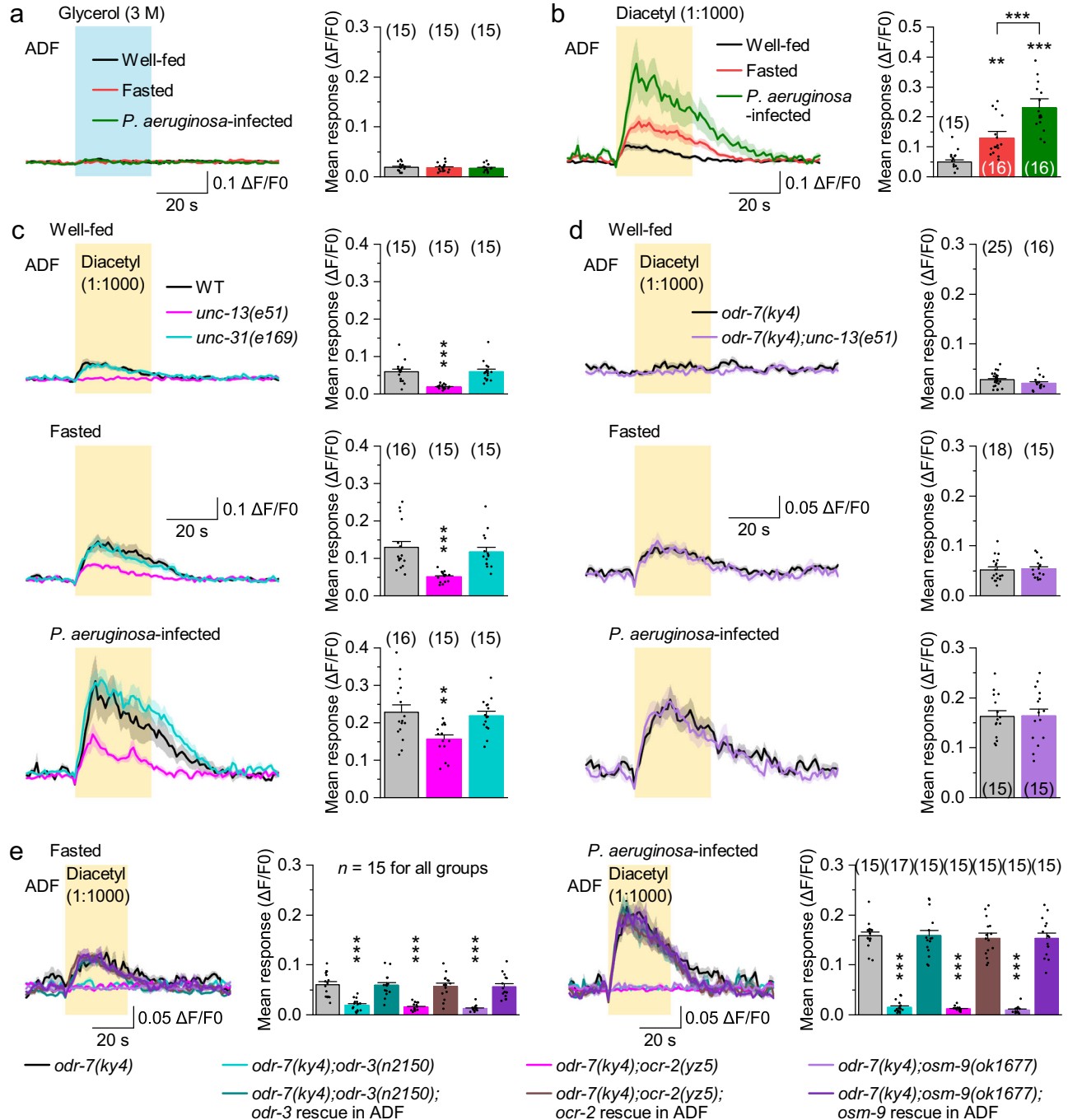

**Fig. 2 | ADF neurons act as primary olfactory sensory neurons for diacetyl in fasted and infected states.** Glycerol-evoked (**a**) and diacetyl-evoked (**b**) GCaMP6 responses in ADF neurons of well-fed, fasted, or *P. aeruginosa*-infected worms. Left, averaged GCaMP6 signals (solid lines, mean; shaded regions, SEM). Right, comparisons of mean responses during the 30-s stimulation period. GCaMP6 was expressed in ADF using *Psrh-142*. $p$ = 0.9152 and 0.7084 (**a**), and 0.007, 0.0001, and 0 (**b**). **c**–**e** Diacetyl-evoked GCaMP6 responses in ADF neurons of worms with the indicated genotypes and conditions. ADF-specific rescue was performed using *Psrh-142*. WT wild type. $p$ = <0.0001, 0.9999, 0.0001, 0.7778, 0.0043, and 0.8963 (**c**), 0.0898, 0.8251, and 0.9601 (**d**), and <0.0001, 1, <0.0001, 0.9989, <0.0001, 0.9943, 0, 1, 0, 0.999, 0, and 0.9989 (**e**). **$p$ < 0.01 and ***$p$ < 0.001 (one-way ANOVA with Tukey's post hoc test for (**a**–**c**, **e**); two-sided unpaired *t*-test for (**d**). Brackets indicate the number of animals tested. Data are shown as means ± SEM. Source data are provided as a Source Data file.

We next examined the role of the PMK-1/p38 MAPK immune pathway, which cooperates with DAF-16 in epithelial immunity against *P. aeruginosa*[48,54]. Given that p38 MAPK can activate AKT and antagonize DAF-16 nuclear translocation[55], we hypothesized that *P. aeruginosa* modulates ADF diacetyl sensitivity via p38-mediated inhibition of DAF-16. Indeed, in infected *odr-7(ky4)* mutants, ADF responses were abolished by mutations or intestine-specific RNAi of p38 MAPK pathway genes *nsy-1* or *sek-1* (Fig. 4c). Unexpectedly, deletion of *pmk-1* had

no effect (Fig. 4c). The *C. elegans* genome encodes three p38 homologs, PMK-1, PMK-2, and PMK-3. Intestine-specific RNAi of *pmk-2* eliminated ADF responses in *odr-7(ky4)* mutants, whereas deletion of *pmk-3* had no effect (Fig. 4c). As previously reported[54], *P. aeruginosa* infection significantly inhibited DAF-16 nuclear translocation (Fig. 4b). This inhibition was reversed by intestine-specific RNAi of *pmk-2* (Fig. 4b). Notably, intestinal DAF-16a^AM overexpression suppressed ADF responses in *odr-7(ky4)* mutants, with no additive effect when

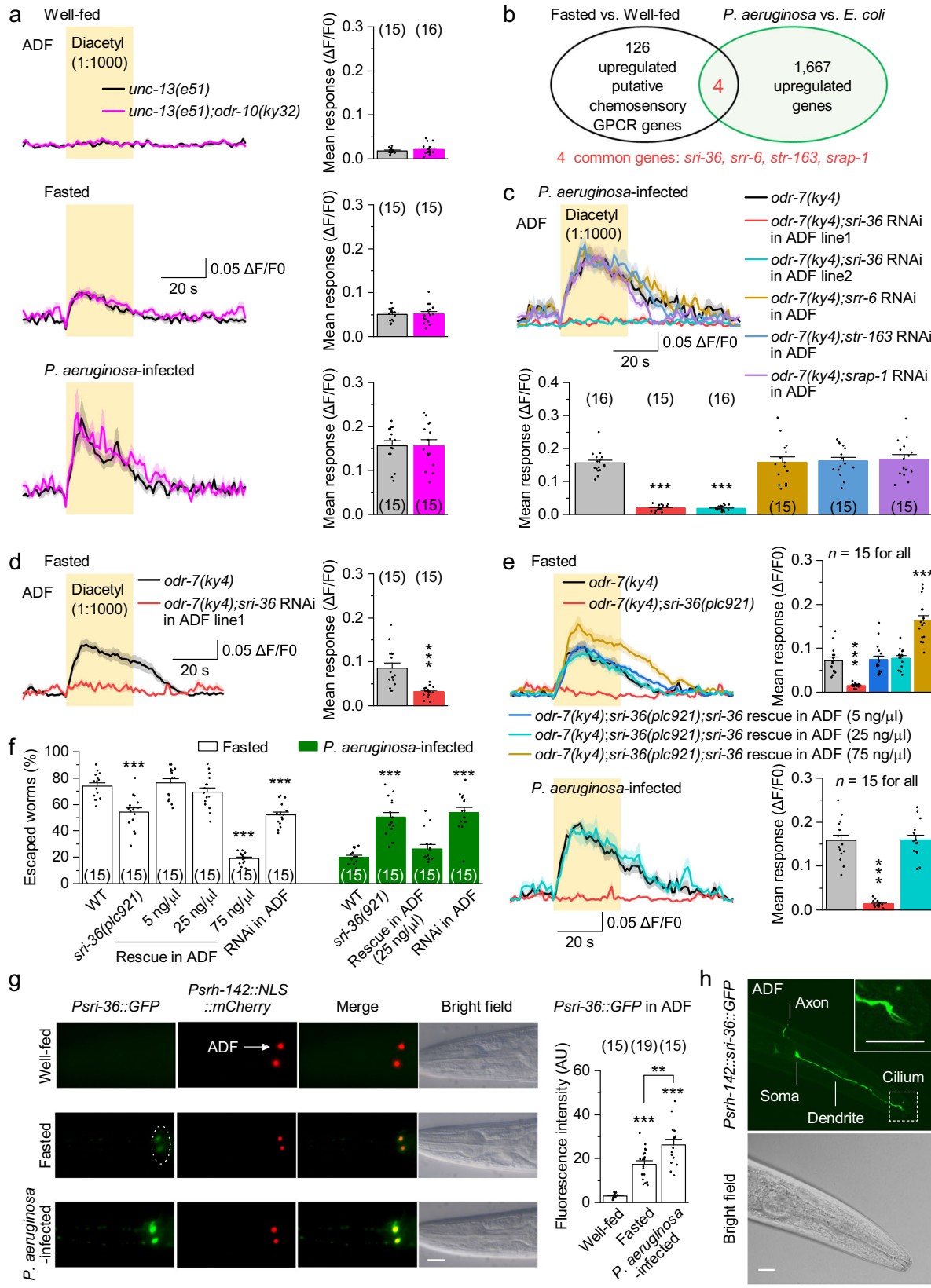

combined with intestine-specific RNAi of *pmk-2* (Fig. 4c). Comparable intestine-specific RNAi effects were observed in the *sid-1(qt9)* background (Fig. 4b, c), confirming tissue specificity. These results suggest that PMK-2 inhibits DAF-16 in IECs to suppress ADF diacetyl sensitivity during *P. aeruginosa* infection.

We then asked whether these pathways regulate *sri-36* expression in ADF and influence decision-making. Intestine-specific RNAi of *aak-2* or *daf-16* similarly reduced *Psri-36::GFP* fluorescence and escape in fasted worms (Fig. 4d, e). Simultaneous RNAi of both genes produced no additive effect on escape (Fig. 4e). In infected worms, intestine-

**Fig. 3 | Hunger- and infection-induced SRI-36 acts as a diacetyl receptor in ADF neurons. a** Diacetyl-evoked GCaMP6 responses in ADF neurons of *unc-13(e51)* or *unc-13(e51);odr-10(ky32)* mutants under the indicated conditions. Left, averaged GCaMP6 signals (solid lines, mean; shaded regions, SEM). Right, comparisons of mean responses during the 30-s stimulation period. $p = 0.4287$, 0.9069, and 0.9945. **b** Transcriptomic analysis of published RNA-sequencing datasets. A Venn diagram shows 126 putative chemosensory GPCR genes upregulated during fasting and 1667 genes upregulated during *P. aeruginosa* infection. Four genes were common: *sri-36, srr-6, str-163,* and *srap-1.* **c–e** Diacetyl-evoked GCaMP6 responses in ADF neurons of worms with the indicated genotypes and conditions. ADF-specific RNAi and rescue were performed using *Psrh-142.* $p = 0$, 0, 1, 0.9985, and 0.9662 (**c**), 0.0001 (**d**), and <0.0001, 1, 0.9906, 0, 0, and 0.9945 (**e**). **f** Escape percentages of

fasted or *P. aeruginosa*-infected worms with the indicated genotypes. WT wild type. $p = $ <0.0001, 0.9903, 0.8333, 0, <0.0001, <0.0001, 0.5497, and 0. **g** *Psri-36::GFP* expression in well-fed, fasted, or *P. aeruginosa*-infected worms. Shown are representative images and comparisons of GFP fluorescence intensity. ADF neurons were labeled with *Psrh-142::NLS::mCherry.* AU arbitrary units. Scale bar, 20 µm. $p = $ <0.0001, 0.0018, and 0. **h** Subcellular localization of the SRI-36::GFP translational fusion protein in ADF neurons. The inset shows GFP signals in the ADF cilium (region with dashed outline). Scale bar, 20 µm. A similar pattern was observed in 16 animals. **p < 0.01 and ***p < 0.001 (one-way ANOVA with Tukey's post hoc test for **c, e–g**; two-sided unpaired *t*-test for **a, d**). Brackets indicate the number of animals tested (**a, c–e, g**) or independent assays (**f**). Data are shown as means ± SEM. Source data are provided as a Source Data file.

specific RNAi of *pmk-2* or overexpression of DAF-16a<sup>AM</sup> similarly reduced *Psri-36::GFP* expression and impaired escape (Fig. 4f, g). RNAi of *pmk-2* in the presence of DAF-16a<sup>AM</sup> also produced no additive effect on escape (Fig. 4g). Comparable intestine-specific RNAi effects were observed in the *sid-1(qt9)* background (Fig. 4d–g), confirming tissue specificity. These results suggest that both the AAK-2/AMPK energy-sensing pathway and the PMK-2/p38 MAPK immune pathway remotely regulate ADF *sri-36* expression via DAF-16/FOXO from the intestine.

### Intestine-released ILPs regulate SRI-36 expression via the IIS pathway in ADF

To investigate how intestinal signals regulate SRI-36 expression in ADF, we considered endocrine signaling, a conserved mechanism for gut-to-brain communication[24,56]. The *C. elegans* genome encodes three major neuropeptide families, FLPs, NLPs, and ILPs[57]. Intestine-specific RNAi of *hid-1*, a gene required for intestinal neuropeptide sorting and secretion[58], or deletion of *egl-3*, encoding a proprotein convertase essential for neuropeptide maturation[57], abolished ADF responses in both fasted and infected *odr-7(ky4)* mutants (Fig. 5a). Comparable intestine-specific RNAi effects were observed in the *sid-1(qt9)* background (Fig. 5a), confirming tissue specificity. Similarly, ADF-specific RNAi of *daf-2*, encoding the sole ILP receptor in *C. elegans*[59], eliminated ADF responses (Fig. 5b). These results suggest that intestinal ILPs act via DAF-2 in ADF to promote diacetyl sensitivity.

The *C. elegans* genome encodes 40 ILPs[57,59]. To identify the specific ILPs involved, we performed a feeding RNAi screen using the intestine-specific RNAi strain VP303 and found that INS-37 and INS-7 selectively regulated ADF responses in fasted and infected worms, respectively (Supplementary Fig. 4). These effects were confirmed by intestine-specific RNAi of *ins-37* or *ins-7* via plasmid injection (Supplementary Fig. 4c, f). Additionally, intestine-specific RNAi of *ins-37* and deletion of *ins-7* abolished ADF responses in fasted and infected *odr-7(ky4)* mutants, respectively (Fig. 5c, d). The defect of *ins-7(tm2001)* was rescued by intestine-specific expression of wild-type *ins-7* and recapitulated by intestine-specific RNAi (Fig. 5d). Furthermore, fasting-induced expression of the transcriptional reporter *Pins-37::GFP*, while *P. aeruginosa* infection suppressed it (Fig. 5e). Intestine-specific RNAi of *aak-2*, *daf-16*, or both genes reduced *Pins-37::GFP* expression similarly in fasted worms (Fig. 5f). In contrast, *P. aeruginosa* infection significantly increased *Pins-7::GFP* expression, which was blocked by intestine-specific RNAi of *pmk-2* or overexpression of DAF-16a<sup>AM</sup>, with no additive effect when combined (Fig. 5g). Comparable intestine-specific RNAi effects were observed in the *sid-1(qt9)* background (Fig. 5c, d, f, g), confirming tissue specificity. Notably, *Pins-7::GFP* expression did not differ significantly between 4 h and 24 h of infection but was enhanced by 0.35% peptone, consistent with escape behavior (Supplementary Fig. 5a, b). Among other tested bacteria, although *S. aureus* also colonized the intestine, it did not induce *Pins-7::GFP* expression (Supplementary Fig. 5e, f). These results are consistent with a model in which hunger and *P. aeruginosa* infection

induce intestinal INS-37 and INS-7 release, respectively, via AAK-2-DAF-16 and PMK-2-DAF-16 pathways, which in turn regulate ADF diacetyl sensitivity.

To test the specificity of these pathways, we performed reciprocal reporter assays. Fasting-induced *Pins-37::GFP expression* was not altered by intestinal *pmk-2* RNAi (Supplementary Fig. 5c). In contrast, *P. aeruginosa* infection-induced *Pins-7::GFP* expression was further enhanced in *aak-2* RNAi worms after 1 h and 2 h of infection, but not at 4 h, suggesting a transient peak of infection-induced *ins-7* expression (Supplementary Fig. 5d). These results indicate that *pmk-2* signaling is not required for fasting-induced *ins-37* expression, whereas *aak-2* normally acts to dampen *ins-7* induction during infection. Thus, while *aak-2* and *pmk-2* serve as primary mediators of fasting- and infection-induced responses, respectively, these pathways also exhibit limited cross-talk.

To determine whether INS-37 and INS-7 regulate SRI-36 expression, we analyzed *Psri-36::GFP* fluorescence in ADF and escape behavior. In fasted worms, intestine-specific *ins-37* RNAi or ADF-specific *daf-2* RNAi similarly reduced *Psri-36::GFP* expression and escape, with no additive effect when combined (Fig. 5h, i). In infected worms, intestine-specific *ins-7* RNAi or ADF-specific *daf-2* RNAi similarly suppressed *Psri-36::GFP* expression and enhanced escape, again with no additive effect when combined (Fig. 5h, i). These results demonstrate that intestine-released INS-37 and INS-7 act through DAF-2 in ADF to regulate SRI-36 expression and decision-making. Consistently, *P. aeruginosa* also induced *Pins-7::GFP* expression and weak *Psri-36::GFP* expression in well-fed worms (Supplementary Fig. 5g, h). However, because *P. aeruginosa* did not affect escape in well-fed worms (Supplementary Fig. 5i), these results suggest that while *P. aeruginosa* activates intestinal immune signaling in both well-fed and fasted worms; its behavioral suppression is most pronounced under fasting, when hunger-driven risk-taking is elevated.

To dissect transcriptional mechanisms downstream of DAF-2, we examined known effectors of the IIS pathway. DAF-16/FOXO, HSF-1/HSF, SKN-1/Nrf2, PHA-4/FOXA, and DAF-12 are generally inhibited by DAF-2, while PQM-1 is activated by DAF-2[59,60]. Interestingly, the upstream regulatory sequence of *sri-36* contains predicted binding sites[60] for both DAF-16 and PQM-1 (Supplementary Fig. 6a). In well-fed *odr-7(ky4)* mutants, ADF-specific RNAi of *daf-16*, but not other IIS effectors, induced robust diacetyl responses in ADF (Supplementary Fig. 6b). In infected worms, ADF-specific RNAi of *pqm-1* had no effect (Supplementary Fig. 6c). These results suggest that DAF-16 represses *sri-36* expression in ADF.

Consistently, ADF-specific *daf-16* RNAi induced *Psri-36::GFP* expression in well-fed worms, while ADF-specific DAF-16a<sup>AM</sup> overexpression suppressed *Psri-36::GFP* expression and impaired escape in both fasted and infected worms (Fig. 5j, k). These results support a model in which DAF-16 acts cell-autonomously in ADF to inhibit *sri-36* expression, and that ILP-DAF-2 signaling relieves this inhibition to drive sensory plasticity during hunger and infection.

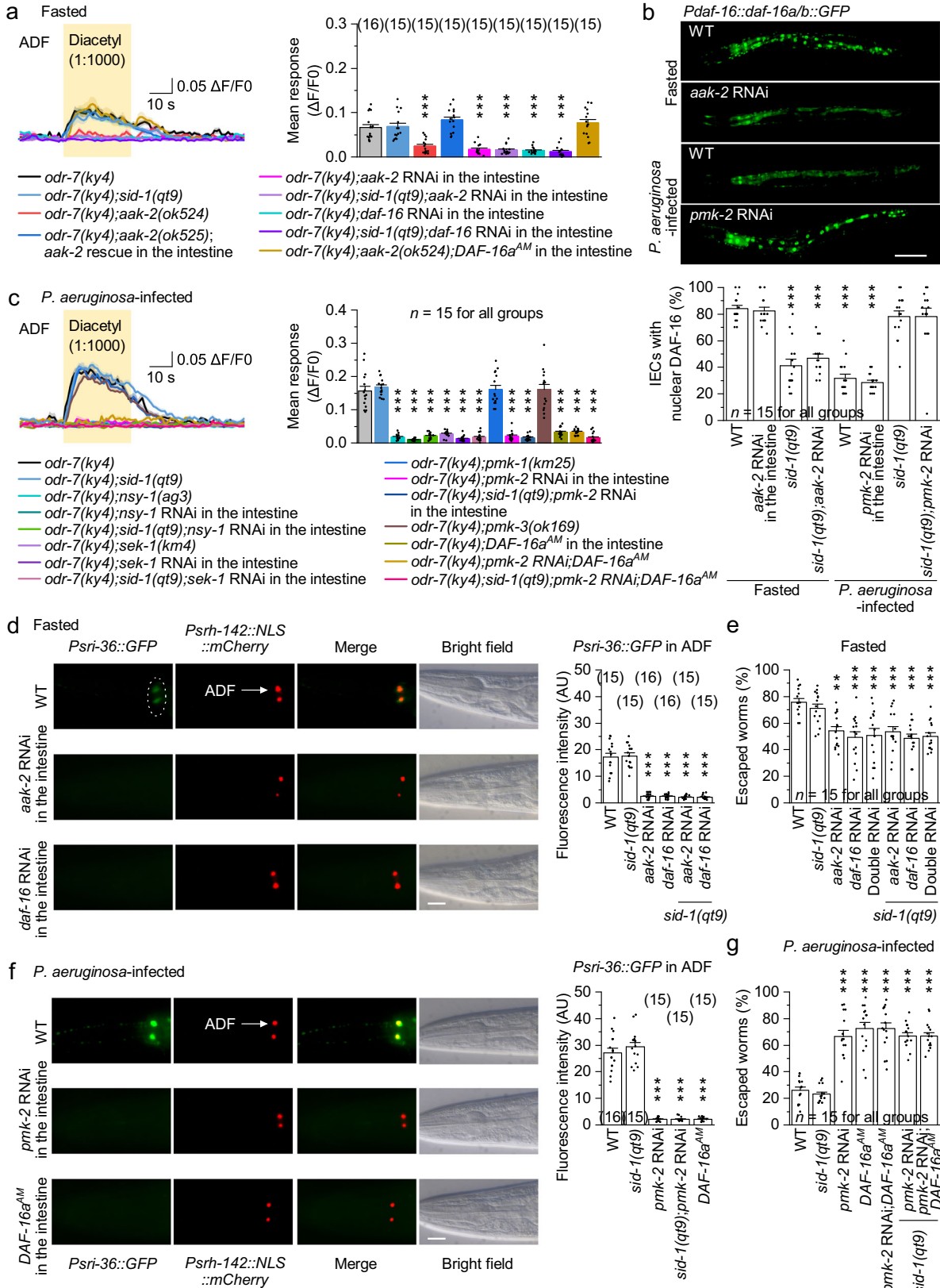

### AIB/AUA and AIA/AIY interneurons act downstream of ADF to regulate behavioral choices

We next sought to identify the interneurons that act downstream of ADF to mediate state-dependent decision-making. ADF neurons form synapses with multiple interneurons, including AIA, AIB, AIY, AIZ, AUA, and RIA[38], with AIA, AIB, AIY, and AIZ serving as first-layer chemosensory interneurons[26]. In fasted worms, silencing AIB alone, AIB/AIZ, or AUA/URX using HisCl1 significantly reduced escape, whereas silencing AIY/AIZ, URX alone, or other interneurons had no effect (Fig. 6a). Notably, simultaneous silencing of AIB/AUA/URX further decreased escape compared to silencing AIB or AUA/URX alone (Fig. 6a). In infected worms, silencing AIA or AIY significantly increased

**Fig. 4 | Hunger- and infection-induced SRI-36 expression in ADF requires intestinal AAK-2/AMPK and PMK-2/p38 MAPK pathways. a, c** Diacetyl-evoked GCaMP6 responses in ADF neurons of worms with the indicated genotypes and conditions. Left, averaged GCaMP6 signals (solid lines, mean; shaded regions, SEM). Right, comparisons of mean responses during the 30-s stimulation period. Intestine-specific expression and RNAi were performed using *Pges-1*. The systemic RNAi-defective *sid-1(qt9)* background was used to abolish dsRNA spreading and validate tissue specificity. $p = 1$, <0.0001, 0.3912, <0.0001, <0.0001, <0.0001, <0.0001, and 0.9025 (**a**), and 0.9979, <0.0001, <0.0001, <0.0001, 0, <0.0001, <0.0001, 1, <0.0001, <0.0001, 1, 0, 0, and <0.0001 (**c**). **b** Nuclear localization of DAF-16a/b::GFP in fasted or *P. aeruginosa*-infected worms with the indicated genotypes. Top, representative images. Bottom: comparisons of the percentage of

GFP-labeled IEC nuclei per animal. WT wild type. Scale bar, 50 μm. $p = 1$, <0.0001, <0.0001, <0.0001, <0.0001, 0.9635, and 0.9635. *Psri-36::GFP* expression in fasted (**d**) or *P. aeruginosa*-infected (**f**) worms with the indicated genotypes. Shown are representative images and comparisons of fluorescence intensity. ADF neurons were labeled with *Psrh-142::NLS::mCherry*. AU arbitrary units. Scale bars, 20 μm. $p = 0.9985$, 0, 0, 0, and 0 (**d**), and 0.5826, <0.0001, <0.0001, and <0.0001 (**f**). Escape percentages of fasted (**e**) or *P. aeruginosa*-infected (**g**) worms with the indicated genotypes. $p = 0.9807$, 0.0013, <0.0001, 0.0001, 0.0008, <0.0001, and <0.0001 (**e**), and 0.9962, <0.0001, 0, 0, <0.0001, and <0.0001 (**g**). **$p < 0.01$ and ***$p < 0.001$ (one-way ANOVA with Tukey's post hoc test). Brackets indicate the number of animals tested (**a–d, f**) or independent assays (**e, g**). Data are shown as means ± SEM. Source data are provided as a Source Data file.

escape, and the effect was amplified when both were silenced simultaneously (Fig. 6a). To investigate how ADF neurons regulate these interneurons, we performed calcium imaging. In fasted *odr-7(ky4)* mutants, diacetyl exposure inhibited AIB and activated AUA (Fig. 6b). In infected mutants, diacetyl inhibited both AIA and AIY (Fig. 6c). Silencing ADF using HisCl1 abolished diacetyl-evoked responses in these interneurons (Fig. 6b, c), whereas silencing AIB/AUA or AIA/AIY had no effect on ADF calcium responses (Supplementary Fig. 7a). These results suggest that ADF neurons regulate decision-making through distinct interneuron pathways depending on internal states. In fasted worms, ADF neurons promote risk-taking by inhibiting AIB and activating AUA, whereas in infected worms, they suppress risk-taking by inhibiting AIA and AIY. This is consistent with established roles of AIB in promoting avoidance, AUA in suppressing avoidance, and AIA/AIY in promoting attraction[26,61].

We next identified the 5-HT receptors mediating ADF's effects on these interneurons. Among the six known *C. elegans* 5-HT receptors[62], deletion of *ser-4* and *lgc-50* significantly reduced escape in fasted worms, while deletion of *mod-1* increased escape in infected worms (Supplementary Fig. 7b). *ser-4*, *lgc-50*, and *mod-1* encode a Gi/o-coupled inhibitory receptor, a 5-HT-gated chloride channel, and a 5-HT-gated cation channel, respectively[62]. Based on the behavioral roles and activity patterns of AIB, AUA, AIA, and AIY (Fig. 6a–c), we hypothesized that SER-4, LGC-50, and MOD-1 function in AIB, AUA, and AIA/AIY, respectively. Indeed, neuron-specific expression of the corresponding wild-type receptor genes in these interneurons rescued the behavioral phenotypes of the respective mutants (Fig. 6d), consistent with their known expression patterns[62]. Furthermore, neuron-specific RNAi of *ser-4*, *lgc-50*, and *mod-1* abolished exogenous 5-HT-evoked calcium responses in AIB, AUA, and AIA/AIY, respectively (Fig. 6e, f). These results identify SER-4, LGC-50, and MOD-1 as key 5-HT receptors that relay ADF 5-HT output to distinct interneurons.

Finally, we tested whether direct modulation of ADF activity is sufficient to shift decision-making. Because diacetyl evoked stronger ADF activation in infected worms than in fasted worms (Fig. 2), we hypothesized that *P. aeruginosa* infection increases ADF 5-HT release. Indeed, infected *odr-7(ky4)* mutants exhibited significantly stronger synapto-pHluorin (SNB-1::pHluorin, a synaptic vesicle release reporter[63]) signals in ADF upon diacetyl stimulation than fasted worms (Fig. 6g), demonstrating enhanced 5-HT release during infection. Consistent with bidirectional control by SRI-36, in fasted *sri-36(plc921)* mutants, low-level SRI-36 expression (5 ng/μl) restored moderate 5-HT release to a level similar to fasted controls, whereas high-level expression (75 ng/μl) enhanced 5-HT release to a level comparable to *P. aeruginosa*-infected controls (Fig. 6g). Chemogenetic manipulation of ADF activity further supported this bidirectional control. Enhancing ADF activity in fasted worms using TRPV1, a mammalian cation channel activated by capsaicin[64], reduced escape in a dose-dependent manner, an effect abolished by AIA/AIY-specific *mod-1* RNAi (Fig. 6h). Conversely, chemogenetically suppressing ADF activity in infected worms using HisCl1 increased escape in a dose-dependent fashion (Fig. 6i). Notable, histamine concentrations above 3 mM reversed this trend

and again suppressed escape, consistent with ADF promoting risk-taking at moderate activity levels but suppressing it when excessively activated. This bidirectional effect was attenuated by RNAi of *ser-4* in AIB and *lgc-50* in AUA (Fig. 6i). Together, these results demonstrate that tuning ADF activity is sufficient to switch behavioral strategies, with moderate ADF activation promoting risk-taking through AIB/AUA and strong activation suppressing it via AIA/AIY.

## Discussion

In this study, we investigate how intestinal bacteria and metabolic states intersect to regulate host decision-making in *C. elegans*. Our findings uncover a gut-to-brain mechanism whereby hunger promotes risk-taking while intestinal *P. aeruginosa* suppresses it, both by modulating central 5-HT signaling. Fasting and infection differentially regulate the chemoreceptor SRI-36 in ADF 5-HT neurons, altering their sensitivity to diacetyl and driving opposing behavioral strategies via distinct downstream interneurons. These sensory modulations are orchestrated by the intestinal AAK-2/AMPK energy-sensing and PMK-2/p38 MAPK immune pathways, which act through a shared FOXO-to-FOXO axis but engage different ILPs (Fig. 7). Together, these findings define a dynamic regulatory interface linking metabolism, immunity, and behavioral plasticity.

The state-dependent sensory modulation enables *C. elegans* to flexibly adapt behavior based on internal states. In natural environments, diacetyl attracts *C. elegans* to rotting fruit, a valuable but risky food source that often harbors pathogens[65]. Hunger increases diacetyl attraction, promoting foraging and survival[3–6], whereas pathogen infection suppresses this attraction, minimizing exposure to harmful microbes. These bidirectional shifts demonstrate how internal states reshape sensory representations and behavioral priorities to support context-appropriate decision-making. Furthermore, our findings indicate that suppression of hunger-driven risk-taking is specific to *P. aeruginosa*, which activates the intestinal PMK-2/p38 MAPK pathway. In contrast, *S. aureus* colonizes the intestine, but neither activates this immune signaling nor alters behavior. These results suggest that the differential effects of pathogens reflect distinct activation of intestinal immune pathways rather than differences in colonization or persistence. Future studies will be important to test additional pathogens that both colonize and activate PMK-2 signaling, to determine whether this mechanism for overriding hunger-driven decision-making is broadly conserved across diverse microbial infections.

Behavioral outputs depend not only on sensory input but also on how neural circuits process and route that input. The same stimuli can elicit distinct behaviors depending on the physiological state of sensory neurons and the downstream circuits they engage[22,66,67]. While such flexibility is essential for adaptive decision-making, the underlying neural architectures have remained elusive. Our study identifies a circuit motif in which ADF 5-HT neurons, modulated by intestinal metabolic and immune cues, act as a state-sensitive switch that adjusts 5-HT output to engage distinct downstream interneuron pathways. This configuration enables the nervous system to switch behavioral

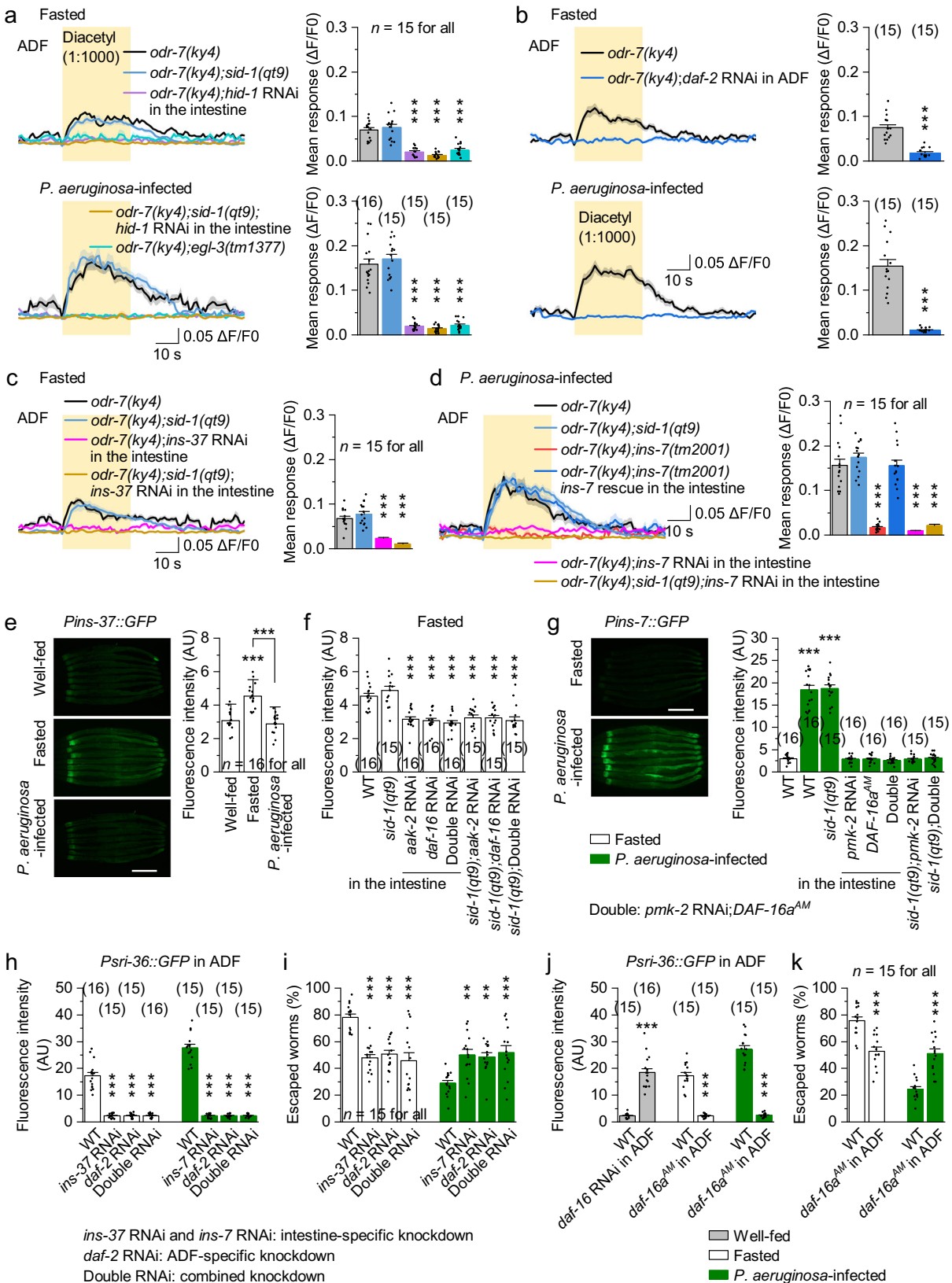

*ins-37* RNAi and *ins-7* RNAi: intestine-specific knockdown
*daf-2* RNAi: ADF-specific knockdown
Double RNAi: combined knockdown

strategies in response to changing internal demands. Notably, fasting still increased escape even when 5-HT synthesis, ADF activity, or *sri-36* was disrupted (Supplementary Fig. 8), suggesting the presence of parallel, 5-HT-independent pathways. These may include other neuromodulators such as tyramine[6], dopamine[68], and additional signals[69] that contribute to hunger-driven risk-taking. Future work will be

needed to delineate these parallel mechanisms and determine how they interact with the 5-HT axis to shape adaptive decision-making.

Central 5-HT is a critical neuromodulator of threat and reward processing and plays a key role in neuropsychiatric disorders associated with impaired decision-making, including autism, schizophrenia, anxiety, and depression[31,70–73]. Despite extensive study, 5-HT's

**Fig. 5 | Intestine-released ILPs regulate SRI-36 expression via the IIS pathway in ADF. a**–**d** Diacetyl-evoked GCaMP6 responses in ADF neurons of worms with the indicated genotypes and conditions. Left, averaged GCaMP6 signals (solid lines, mean; shaded regions, SEM). Right, comparisons of mean responses during the 30-s stimulation period. Intestine- and ADF-specific RNAi were performed using *Pges-1* and *Psrh-142*, respectively. The *sid-1(qt9)* background was used to validate tissue specificity. $p = 0.8812, 0, 0, <0.0001, 0.8016, 0, 0,$ and 0 (**a**), $<0.0001$ and $<0.0001$ (**b**), $0.3917, <0.0001,$ and 0 (**c**), and $0.6576, 0, 1, 0,$ and 0 (**d**). **e, f** *Pins-37::GFP* expression in worms with the indicated genotypes and conditions. Shown are representative images and comparisons of fluorescence intensity. Intestine-specific RNAi was performed using *Pges-1*. WT wild type. AU arbitrary units. Scale bar, 100 µm. $p = <0.0001, 0,$ and 0.7061 (**e**), and $0.8978, <0.0001, <0.0001,$

$<0.0001, <0.0001, <0.0001,$ and $<0.0001$ (**f**). **g** *Pins-7::GFP* expression in fasted or *P. aeruginosa*-infected worms with the indicated genotypes. Shown are representative images and comparisons of fluorescence intensity. Scale bar, 100 µm. $p = <0.0001, <0.0001, 1, 1, 0.9973, 1,$ and 1. **h, j** Comparisons of *Psri-36::GFP* fluorescence intensity in worms with the indicated genotypes and conditions. $p = 0, 0, 0, <0.0001, <0.0001,$ and $<0.0001$ (**h**), and $<0.0001, <0.0001,$ and $<0.0001$ (**j**). **i, k** Escape percentages of worms with the indicated genotypes and conditions. $p = <0.0001, <0.0001, <0.0001, 0.0011, 0.0025,$ and 0.0004 (**i**), and $<0.0001$ and $<0.0001$ (**k**). **\*\****p < 0.01$ and **\*\*\****p < 0.001$ (one-way ANOVA with Tukey's post hoc test for **a, c**–**i**; two-sided unpaired *t*-test for (**b, j, k**). Brackets indicate the number of animals tested (**a**–**h, j**) or independent assays (**i, k**). Data are shown as means ± SEM. Source data are provided as a Source Data file.

function remains paradoxical. It facilitates aversive learning and threat detection but also modulates reward processing, with dysregulation contributing to anhedonia[2,16,70,72,74–76]. Clinically, selective serotonin reuptake inhibitors (SSRIs) remain first-line antidepressants, yet they often fail to relieve, and may even exacerbate, motivational symptoms and anhedonia[77]. The heterogeneous nature of 5-HT neurons, their widespread brain projections, and diverse receptors complicates efforts to resolve these opposing effects[62,71,78]. Our findings suggest a unifying model in which central 5-HT acts as a behavioral rheostat, promoting risk-taking at moderate levels and suppressing it when excessively released.

We further show that internal states regulate 5-HT output by modulating chemoreceptor expression in ADF neurons. This sensory plasticity provides a mechanism for flexible behavioral responses to the same external stimulus[8,29,79]. Building on prior work showing that fasting and infection can modulate chemoreceptor expression through insulin signaling[29,47,80,81], our study identifies intestine-released ILPs as key molecular signals linking interoception to sensory gain control. Specifically, fasting-induced INS-37 and pathogen-induced INS-7 engage the canonical insulin signaling pathway to regulate SRI-36 expression in ADF. This chemoreceptor remodeling tunes ADF diacetyl sensitivity and, consequently, the level of 5-HT release, thereby driving divergent food-seeking behaviors. Notably, chemoreceptors are often encoded in large gene families in many animals[46], potentially providing a flexible repertoire that can be selectively induced or suppressed by physiological state. We propose that such plasticity enables animals to integrate complex internal and environmental cues to generate coherent behavioral responses under dynamic conditions. In this context, SRI-36 modulation offers a model for how gut-derived cues reshape sensory input to support state-dependent behavioral strategies. These findings suggest that olfactory tuning may be a promising target for treating metabolic disorders[82].

This ILP-based mechanism of sensory modulation may be evolutionarily conserved. In mammals, ILP receptors are widely expressed in the brain, including olfactory regions[82,83], and gut-derived ILPs can cross the blood-brain barrier[83], suggesting that ILPs may regulate neural function beyond their metabolic effects. Indeed, ILPs influence neurotransmitter release, receptor expression, and neuronal excitability[83] and are critical regulators of feeding-related behaviors[7,83]. For example, central insulin administration improves odor discrimination and reduces feeding[7,83], while in *Drosophila*, ILPs modulate olfactory neuron sensitivity and increase aversion to noxious food under starvation[7,84]. Moreover, subsets of IECs can detect pathogens via innate immune responses and release ILPs[85]. Although most mammalian ILPs remain uncharacterized, gut-derived ILPs such as INSL5 are upregulated by fasting and influenced by the microbiota[86]. These observations raise the possibility that gut-derived ILPs are conserved mediators of sensory plasticity and behavioral adaptation.

We also show how a single sensory neuron, ADF, can flexibly reconfigure behavior by engaging distinct downstream circuits. In the fasted state, moderate ADF activation biases behavior toward food

attraction by inhibiting AIB and activating AUA. In the infected state, heightened ADF activation suppresses attraction by inhibiting AIA and AIY. These opposing effects are mediated by distinct 5-HT receptors, SER-4, LGC-50, and MOD-1, expressed in the respective interneurons, in line with their known behavioral roles[26,61]. This circuit architecture allows *C. elegans* to dynamically switch behavioral strategies based on internal state, highlighting how physiologically tuned sensory neurons coordinate antagonistic behavioral outputs. Future work should quantify 5-HT dose-response effects on downstream interneurons (e.g., changes in speed or turning dynamics across internal states) to further refine this circuit model.

Together, our study advances understanding of how gut microbes influence brain function, an area where mechanisms remain largely undefined[12–15]. We demonstrate that *P. aeruginosa* activates an intestinal immune pathway that reprograms sensory function and overrides hunger-driven decision-making via central 5-HT signaling, pointing to an immune-mind connection. Similar principles may operate in mammals.

A limitation of this study is the use of RNAi to infer cell-specific gene function. Previous studies have shown that dsRNA can spread between tissues[87], and therefore, low-level silencing in unintended cells cannot be excluded. However, RNAi-based results were interpreted in conjunction with complementary approaches, including genetic mutants, tissue-specific rescue, CRISPR knock-in reporters, and functional imaging and behavioral analyses.

# Method

## Bacterial strains and culture

*E. coli* OP50, *P. aeruginosa* PA14, *S. aureus* NCTC8325, and EPEC were cultured under conditions optimized for their growth and virulence[88,89]. *E. coli*, *P. aeruginosa*, and EPEC were grown in LB medium, while *S. aureus* was cultured in tryptic soy broth (Hopebio) supplemented with 10 µg/ml nalidixic acid (Macklin) to optimize growth and prevent overgrowth of contaminants. *P. aeruginosa* and *S. aureus* cultures were maintained under gentle shaking. All bacteria were grown overnight at 37 °C before seeding onto agar plates. *E. coli* was plated on nematode growth medium (NGM), *P. aeruginosa* on NGM supplemented with 0.25% peptone (Aobox) to support robust virulence factor production while maintaining animal viability for behavioral assays. *S. aureus* cultures were diluted 1:5 before plating on tryptic soy agar (Hopebio) containing 10 µg/ml nalidixic acid (Macklin). EPEC was plated on modified NGM containing 2 mg/ml tryptophan (Aladdin) to enhance its virulence[89]. Plates seeded with *E. coli* and EPEC were incubated at 37 °C for 20 h, *P. aeruginosa* plates at 37 °C for 24 h followed by 6 h at 25 °C, and *S. aureus* plates at 37 °C for 6–8 h. All plates were equilibrated at 25 °C for 1–2 h before use. The strains used in this study are listed in Supplementary Data 1.

## *C. elegans* strains and culture

*C. elegans* strains were raised on NGM plates seeded with *E. coli* OP50 at 22 °C. Transgenic lines were generated by microinjection of plasmid

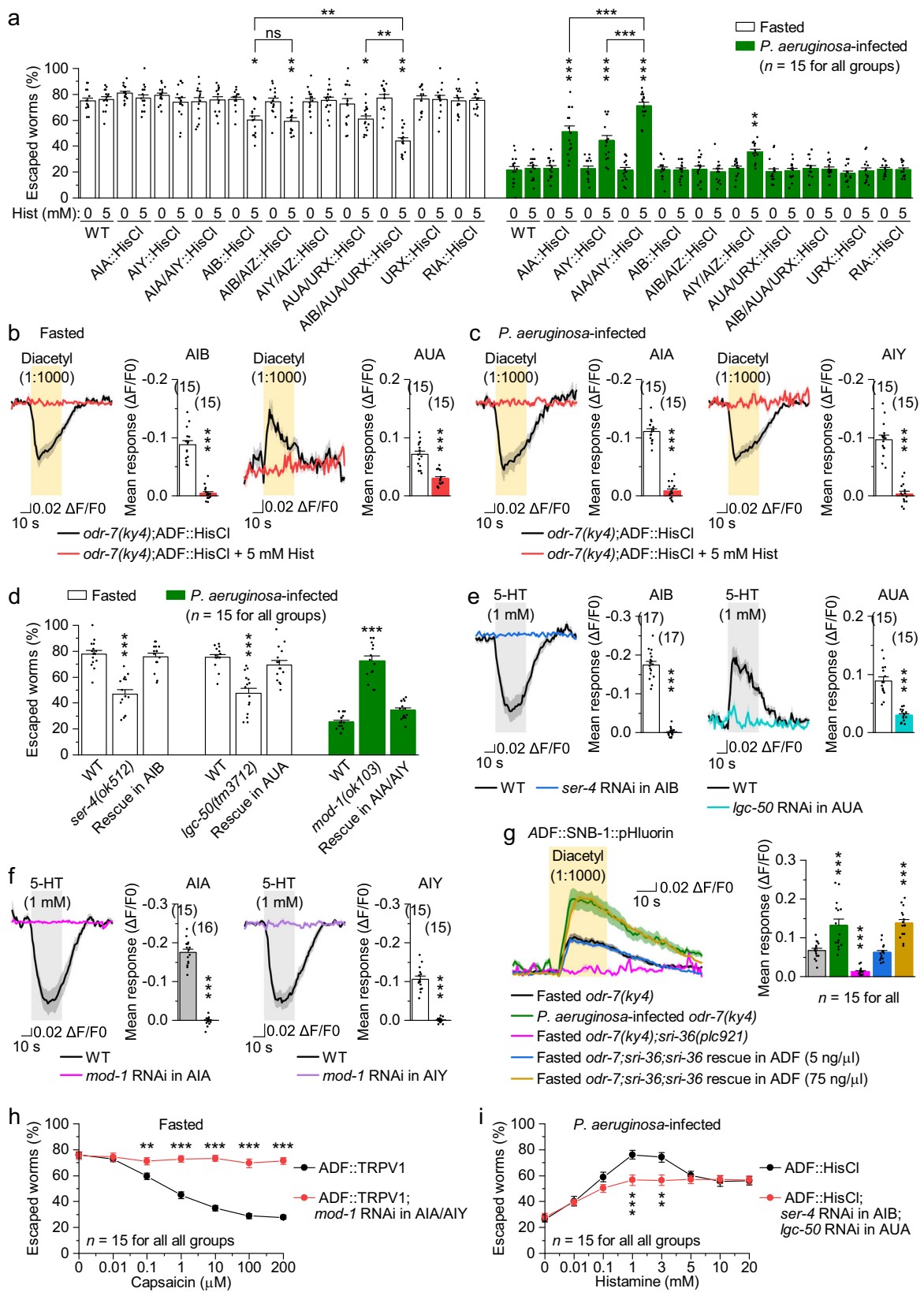

DNA, and at least two independent lines were tested for confirmation. Integrated transgenic strains were created using gamma irradiation and outcrossed at least three times before use. Unless otherwise noted, all experiments used young adult hermaphrodites. Investigators were not blinded to genotype or condition. The strains used in this study are listed in Supplementary Data 1.

## Mutant rescue and RNAi

Mutant rescue was achieved by expressing wild-type cDNAs under cell-specific promoters. RNAi knockdown was performed by co-injecting two plasmids encoding complementary sense and antisense mRNA fragments of the target gene. To confirm tissue specificity and exclude systemic RNAi effects, intestine-specific RNAi experiments (*aak-2*,

**Fig. 6 | AIB/AUA and AIA/AIY interneurons act downstream of ADF to regulate behavioral choices. a** Escape percentages of fasted or *P. aeruginosa*-infected worms with or without 5 mM histamine (Hist) treatment. Neuron-specific HisCl1 expression was performed using *Pgcy-28* (AIA), *Pttx-3* (AIY), *Pnpr-9* (AIB), *Podr-2* (AIB/AIZ), *Pser-2prom2* (AIY/AIZ), *Pflp-8* (AUA/URX), *Pgcy-36* (URX), and *Pglr-3* (RIA). WT wild type. *p* = 1, 0.9958, 1, 1, 1, 1, 1, 0.0137, 1, 0.0059, 1, 1, 1, 0.029, 1, 0, 1, 1, 1, 1, 1, 1, <0.0001, 1, 0, 1, <0.0001, 1, 1, 1, 1, 1, 0.0043, 1, 1, 1, 1, 1, 1, 1, and 1. Diacetyl-evoked GCaMP6 responses in AIB, AUA, AIA, and AIY neurons of fasted (**b**) or *P. aeruginosa*-infected (**c**) worms with the indicated genotypes. Left, averaged GCaMP6 signals (solid lines, mean; shaded regions, SEM). Right, comparisons of mean responses during the 30-s stimulation period. GCaMP6 was expressed in AIB, AUA, AIA, and AIY using *Pnpr-9*, *Pflp-8*, *Pgcy-28*, and *Pttx-3*, respectively. HisCl1 was expressed in ADF using *Psrh-142*. *p* = <0.0001 and <0.0001 (**b**), and <0.0001 and <0.0001 (**c**). **d** Escape percentages of fasted or *P. aeruginosa*-infected worms with the indicated genotypes. Neuron-specific rescue was performed using *Pgcy-28* (AIA), *Pttx-3* (AIY), *Pnpr-9* (AIB), and *Pflp-8* (AUA). *p* = 0, 0.8648, <0.0001, 0.4113, 0, and 0.0384. **e, f** 5-HT-evoked GCaMP6 responses in AIB, AUA, AIA, and AIY neurons of well-fed worms

with the indicated genotypes. Neuron-specific RNAi was performed using *Pgcy-28* (AIA), *Pttx-3* (AIY), *Pnpr-9* (AIB), and *Pflp-8* (AUA). 5-HT was applied directly to the neuron cell body. *p* = <0.0001 and <0.0001 (**e**), and <0.0001 and <0.0001 (**f**). **g** Diacetyl-induced SNB-1:pHluorin fluorescence changes in ADF neurons of fasted or *P. aeruginosa*-infected worms with the indicated genotypes. Left, averaged pHluorin signals (solid lines, mean; shaded regions, SEM). Right, comparisons of mean responses during the 30-s stimulation period. SNB-1::pHluorin was expressed in ADF using *Psrh-142*. *p* = <0.0001, 0.0008, 0.9954, and <0.0001. Escape percentages of fasted or *P. aeruginosa*-infected worms with the indicated genotypes treated with capsaicin (**h**) or histamine (**i**). TRPV1 and HisCl1 were expressed in ADF using *Psrh-142*. *p* = 0.9303, 0.7527, 0.0044, <0.0001, <0.0001, <0.0001, and <0.0001 (**h**), and 0.5764, 0.8202, 0.0775, 0.0007, 0.0033, 0.5149, 0.82, 0.9005 (**i**). **p* < 0.05, ***p* < 0.01, and ****p* < 0.001 (one-way ANOVA with Tukey's post hoc test for **a**, **d**, **g**; two-sided unpaired *t*-test for **b**, **c**, **e**, **f**, **h**, **i**). ns no significance. Brackets indicate the number of independent assays (**a**, **d**, **h**, **i**) or animals tested (**b**, **c**, **e**–**g**). Data are shown as means ± SEM. Source data are provided as a Source Data file.

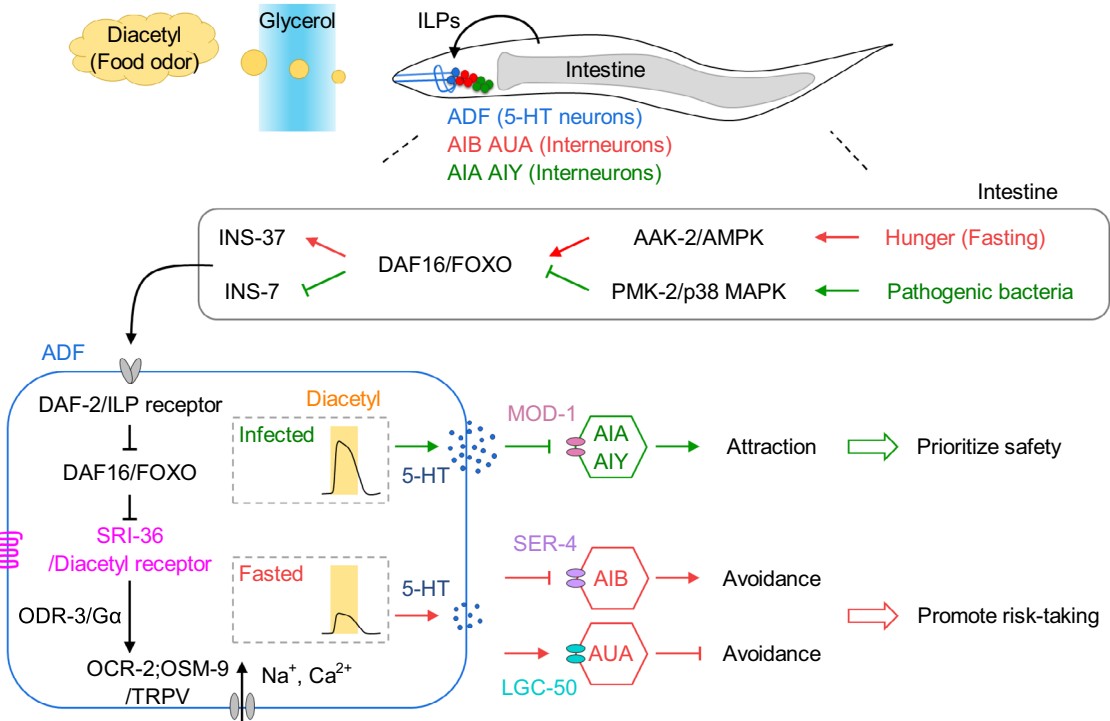

**Fig. 7 | A gut-to-neuron mechanism through which hunger promotes risky decision-making and intestinal pathogens reverse it, both by modulating 5-HT signaling.** Hunger and intestinal pathogens differentially regulate expression of the chemoreceptor SRI-36 in ADF 5-HT neurons, altering their sensitivity to diacetyl and driving opposing behavioral strategies via distinct downstream interneurons. These effects are mediated by the intestinal AAK-2/AMPK energy-sensing and PMK-2/p38 MAPK immune pathways, both acting through a shared FOXO-to-FOXO signaling axis but engaging distinct intestine-released ILPs.

---

*daf-16, nsy-1, sek-1, pmk-2, hid-1, ins-37,* and *ins-7*) were repeated in the systemic RNAi-defective *sid-1(qt9)* mutant background, in which dsRNA spreading is abolished[53]. Promoters used included *Pges-1* (IECs)[49], *Pceh-2* (NSM)[90], *Psrh-142* (ADF)[90], P*clh-3* (HSN)[91], *Pgcy-28* (AIA)[92], *Pttx-3* (AIY)[92], *Pnpr-9* (AIB)[92], *Podr-2* (AIB/AIZ)[28], *Pser-2prom2* (AIY/AIZ)[62], *Pflp-8* (AUA/URX)[93], *Pgcy-36* (URX)[93], *Pglr-3* (RIA)[92], *Psra-6* (ASH)[94], and *Pgpa-6* (AWA)[95]. All cDNAs and RNAi target sequences were cloned from a Bristol N2 cDNA library. Primers (Tsingke Biotechnology Co., Ltd.) are listed in Supplementary Table 1.

## CRISPR-Cas9 genome editing
To generate the *sri-36(plc921)* deletion allele, two sgRNAs (5′ region: AGTACCTGATAAGCCAGACA; 3′ region: TATAATACCAAGAGTGGCCC) were cloned into the pDD162 (*Peft-3::Cas9* + empty sgRNA) vector. The resulting plasmids (20 ng/μl each) were co-injected into young adult hermaphrodites together with *Pmyo-2::mStrawberry* (25 ng/μl) as a co-

injection marker. F1 progeny were screened by PCR for deletion events, and homozygous mutants were confirmed by sequencing. To generate the endogenous *sri-36::mNeonGreen* knock-in reporter, three sgRNAs (GAAATTTGTAATCAGAAAGT; TAACTTTTACAATCCCATAT; ACTCCAT TCGTCTGTCAATG) targeting the *sri-36* C-terminus were used together with a donor repair plasmid containing the *mNeonGreen* coding sequence flanked by ~1 kb homology arms. The injection mixture contained 5 ng/μl sgRNA-Cas9 plasmids, 25 ng/μl donor plasmid, and 25 ng/μl *Pmyo-2::mCherry* co-injection marker. Correct knock-in events were identified by PCR and verified by sequencing. All CRISPR-edited strains were outcrossed three times before use.

## RNAi screen of intestine-released ILPs
Intestine-specific RNAi of ILP genes was performed by feeding *C. elegans* strain VP303 with *E. coli* HT115 carrying L4440 plasmids containing target gene fragments[96]. Briefly, HT115 was cultured overnight

in LB with 100 µg/ml ampicillin (Biosharp) at 37 °C, seeded onto NGM RNAi plates supplemented with 100 µg/ml ampicillin and 5 mM IPTG (Biotopped), and incubated at 37 °C for 12–14 h. Three L4-stage VP303 worms were transferred to each plate at 20 °C until their progeny reached young adulthood. *rab-5* RNAi was used as a positive control.

### Behavioral analysis

All behavioral assays were conducted at 22 °C. Spontaneous locomotion was recorded and analyzed using the *Track-A-Worm* automated tracking system (version 2.0)[23]. A single young adult was transferred to the center of a 6-cm NGM plate, allowed to recover for 30 s, and recorded for 1 min at 15fps.

For behavioral choice assays[23], 6-cm choice plates were prepared by drawing a 1-cm glycerol ring (10 µl, 3 M) and placing two diacetyl spots (1 µl of 1:1000 diacetyl mixed with 1 µl of 0.1 M NaN$_3$) near the plate edges. Synchronized worms were cultured on *E. coli* OP50 until young adulthood and then transferred to plates seeded with different bacterial strains. After 4 h, worms were washed with M9 buffer, and 10–15 worms were placed at the center of each choice plate. The number of worms at the diacetyl spots was counted after 15 min. For fasting assays, worms were deprived of food on unseeded NGM plates for the indicated time before the choice test. NGM plates containing histamine (Sigma-Aldrich) or capsaicin (Aladdin) were prepared by adding the chemicals at their final concentrations, and then seeded with 200 µl of bacteria pre-mixed with the same concentration of histamine or capsaicin.

Unisensory avoidance assays were performed using glycerol-only plates, and worms outside the glycerol ring were counted after 15 min[23]. Unisensory attraction assays were performed by dividing plates into four equal quadrants (A–D)[23]. Diacetyl (1 µl, 1:1000) and control water (1 µl) were applied to opposite edges along with NaN$_3$. Worms were transferred to the center and counted after 15 min. Chemotaxis index = (worms in A − worms in D)/total worms × 100%.

For two-choice chemotaxis quadrant assays[29], 9-cm quadrant plates were divided into four sections, with opposite quadrants containing either diacetyl (20 µl, 1:1000) or control water (20 µl). 100–150 M9-washed worms were placed in the center. After 15 min, plates were frozen at −20 °C for 10 min to stop movement, and worms in diacetyl and control quadrants were counted. Chemotaxis index = (worms in diacetyl − worms in control)/total worms × 100%. A positive index indicates attraction to diacetyl.

### Imaging

Images of the SRI-36::GFP translational fusion and the CRISPR knock-in SRI-36::mNeonGreen fusion protein were acquired using an Olympus IXplore SpinSR confocal microscope with a 100×/1.45 NA oil objective. All other fluorescence images were captured with a Nikon Ts2R inverted microscope equipped with an Mshot MD60 CCD camera and analyzed using the Mshot Image Analysis System (version 1.1). Acquisition settings were kept constant within each experiment. For DAF-16 subcellular localization analysis, strains co-expressing *Pdaf-16::daf-16a/b::GFP* and the IEC nuclear marker *Pges-1::NLS::mCherry* were used. The percentage of GFP-labeled IEC nuclei was counted per animal. For calcium and pHluorin imaging, young adults were immobilized on coverslips using Vetbond Tissue Adhesive (3M Company) in bath solution (140 mM NaCl, 5 mM KCl, 5 mM CaCl$_2$, 5 mM MgCl$_2$, 11 mM dextrose, 5 mM HEPES, pH 7.2, 320 mOsm) and imaged at 1fps for 2 min. Diacetyl or glycerol was pressure-ejected (2 psi, 30 s) near the nose using an Eppendorf FemtoJet 4i microinjector. For exogenous 5-HT stimulation, because the targeted interneurons (AIA, AIY, AIB, and AUA) are located deep within the head and inaccessible to bath-applied 5-HT, a longitudinal incision was made along the glued region to expose neurons. 5-HT was then applied directly to neuron somas using pressure ejections (2 psi, 30 s), ensuring spatial precision, reproducibility, and consistent stimulation of the relevant interneurons.

### Quantification of intestinal bacteria accumulation

To quantify the intestinal accumulation of PA14-dsRed[97], full-lawn plates were prepared by spreading an overnight culture of PA14-dsRed. Young adult hermaphrodites grown on *E. coli* OP50 were transferred onto these plates and exposed for 4 h, then deprived of food on unseeded NGM plates for 6 h. Worms were washed with M9 buffer to remove external bacteria and imaged using the Nikon Ts2R inverted microscope.

Intestinal accumulation of *E. coli, S. aureus*, EPEC, and PA14-*gacA(−)* was quantified using colony-forming unit (CFU) assays[48]. Worms were similarly exposed to bacterial lawns and food-deprived conditions as above. Fifty worms were then collected into M9 containing 0.1 M sodium azide for paralysis, washed three times with M9 containing 1 mg/ml ampicillin (Biosharp) and 1 mg/ml gentamicin (Biosharp), and incubated for 30 min in antibiotic M9 to remove external bacteria. Worms were then lysed with a motorized pestle, serially diluted, and spread on the appropriate agar plates for each bacterial species. Plates were incubated overnight at 37 °C to determine CFUs.

### Data analysis

Fluorescence, calcium, and pHluorin imaging data were analyzed using ImageJ (FIJI, version 1.50i). For neuronal imaging, individual cells were designated as separate regions of interest (ROIs). For the CRISPR knock-in SRI-36::mNeonGreen imaging, the dendritic region was selected as the ROI. For pHluorin imaging, the axonal region in the nerve ring was selected as the ROI. For intestinal fluorescence analysis, a single worm was assigned as the ROI. Calcium and pHluorin traces were first plotted as absolute fluorescence intensity (F) over time and then converted to bleach-corrected ΔF/F0 values using a custom MATLAB module[94].

Statistical analyses and data visualization were performed using Origin 2019 (OriginLab). All diagram images were created using Microsoft PowerPoint 2024. Statistical significance was determined using one-way ANOVA with Tukey's post hoc test for multiple group comparisons and two-sided unpaired *t*-test for two-group comparisons. Detailed statistical methods, sample sizes (*n*), and *p*-values are provided in the figure legends. Statistical significance was set at $p < 0.05$. No data were excluded from analysis, and all data are shown as mean ± SEM.

### Reporting summary

Further information on research design is available in the Nature Portfolio Reporting Summary linked to this article.

## Data availability

All data generated or analyzed during this study are included in this published article and its supplementary information files. Source data are provided with this paper.

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

## Acknowledgements

We would like to thank Jianke Gong, Bin Qi, Anbing Shi, and Taihong Wu for providing bacterial strains; Jianke Gong, Long Lin, Anbing Shi, Haijun Tu, Wenxing Yang, and Donglei Zhang for plasmids; and Cornelia I. Bargmann, Jianke Gong, Long Lin, and Yun Zhang for *C. elegans* strains. We also acknowledge the Caenorhabditis Genetics Center (USA), which is funded by NIH Office of Research Infrastructure Programs (P40OD010440), for providing *E. coli* OP50 and *C. elegans* strains. This work was supported by grants from the National Natural Science Foundation of China (32571188 and 32171003 to P.L.) and the Interdisciplinary Research Program of HUST (5003170102 to P.L.).

## Author contributions

Y.L., C.C., and P.L. conceived and initiated the study. Y.L. and P.L. designed the experiments. Y.L., X.Z., M.X., Y.W., H.L., and J.Z. performed the experiments and analyzed the data. Y.L. and P.L. interpreted the results. P.L. wrote the manuscript.

## Competing interests

The authors declare no competing interests.
