## [Transparent Peer Review file · Nature Communications]

Intestinal pathogens override hunger-driven decision-making via immune regulation of central serotonin signaling in *C. elegans*

Corresponding Author: Dr Ping Liu

Version 0:

Reviewer comments:

Reviewer #1

(Remarks to the Author)

Lei et al. present a study that examines the role of gut-to-brain signaling in the simple animal *C. elegans* in modulating decision-making and behavioral responses to bacterial food cues and pathogenicity. The manuscript gets off to a strong start. The authors set up a nice assay that assesses the “risk-taking” behavior in crossing a noxious chemical barrier to access a food signal. The authors then examine how prior exposure to nutritious bacteria as well as three pathogenic bacteria, and also the effect of bacterial food removal, influence this decision making. The authors carry out a number of solid control experiments and confirm that fasting has a marked effect on subsequent behavioral responses. The authors find that exposure to pathogenic *Pseudomonas aeruginosa* reduces the animal’s risk-taking food-seeking behavior. Genetic experiments with *tph-1* suggest a role for TPH-1 and the ADF neurons in mediating this response. The authors then conduct calcium-imaging experiments of ADF responses to show how ADF activity is influenced during conditions of fasting and *Pseudomonas* exposure. The authors use a candidate expression approach to identify a putative receptor, SRI-36, that is strongly induced in the ADF neurons by fasting and even more so by *Pseudomonas* infection. The authors show that ADF responses are depending on AAK-2, DAF-16, and the NSY-1-SEK-1 p38 pathway. Up until this point, the manuscript is quite impressive with a combination of beautiful genetics and imaging.

However, once the manuscript begins to address site-of-action issues, there are serious concerns with the underlying method being used. The authors claim to be carrying out “cell-specific RNAi.” Methods are somewhat cursory, but it appears that the primary method being utilized is to use cell- or tissue-specific promoters to express + and – strands of a gene of interest off of a plasmid that has been introduced into cells of interest by the generation of transgenic strains. What is not taken into account is that this method does not take into account the ability for dsRNA to spread from cell-to-cell. For this reason, cell-specific RNAi methods are generally performed in an RNAi-defective strain with cell-specific rescue of RNAi (e.g., *rde-1*). The VP303 strain used for the screen for intestine-specific insulins has these attributes. But none of the other ostensibly cell-specific RNAi experiments utilize this strain or have any indication of a cell-specific rescue of RNAi activity. Thus, one cannot conclude anything about cell necessity from these site-of-action experiments. (It appears that the author used this method in a prior 2020 Nature Communications article as well.)

Unless there is a trivial explanation that I am missing, this basic experimental oversight seriously undermines the confidence of this reviewer in the basic methodology and rigor of the manuscript. This is especially disappointing as the first part of the manuscript is remarkably impressive.

Reviewer #2

(Remarks to the Author)

“Intestinal pathogens override hunger-driven decision-making via immunoregulation of central serotonin signaling in *C. elegans*”

Serotonin is known to react to various environmental stresses and internal states to modulate neural functions and generate

behavioral responses. Some of these behavioral responses are orthogonal when the conditions are presented alone to the animals. How serotonin signaling regulates neural activity to produce coherent and balanced behavioral decisions under conditions demanding conflicting outputs from the nervous system is a fundamentally important but challenging question. In the above manuscript, Lei and Chen et al. set out to address this issue.

The authors employed a risk-taking decision paradigm in *C. elegans* where the worms' attraction to an attractive odorant diacetyl (sensed by AWA sensory neuron) is hindered by a hyperosmotic glycerol barrier (sensed by ASH sensory neuron). They characterized how serotonin signaling modulated this decision-making process under two well-controlled conditions. While starvation promotes risk-taking as previously shown, the authors found that feeding on a pathogenic bacteria *Pseudomonas aeruginosa* PA14 suppressed the effect of starvation and inhibited the decision to cross the hyperosmotic barrier.

Using this assay together with cell-specific manipulation of serotonin production and neural activity, the authors showed that starvation and *P. aeruginosa* infection converged on the serotonin signal produced by ADF neurons to regulate behavior decision on diacetyl/glycerol response, and that starvation and infection both induced ADF response to diacetyl. Using previously published expression profiles, RNAi, and calcium imaging, CRISPR and cell-specific rescue, the authors further showed that starvation and *P. aeruginosa* infection upregulated the expression of a GPCR *sri-36* in ADF and that *sri-36* mediated diacetyl responses in fasted and infected worm in ADF neurons and in behavioral decision. The authors further identified two intestinal pathways, the *aak-2/daf-16* pathway and *pmk-2/daf-16* pathway, that upregulated *sri-36* under fasting and infection conditions. These pathways upregulated expression of two insulin-like peptides, *ins-37* and *ins-7*, in fasted and infected worms to modulate ADF expression of *sri-36* and the behavioral decision between diacetyl/glycerol. Furthermore, the authors mapped the downstream neural circuit and serotonin receptors that mediated starvation and infection-regulated diacetyl response in ADF and behavioral decision, and demonstrated that a stronger release of serotonin from ADF in infected worms compared with starved worms suppressed the risk-taking effect of starvation. Based on these findings, they delineate a molecular strategy in a neuronal circuit that integrates food availability and infection to produce a balanced behavior decision between safety-seeking or risk-taking. The authors fully leverage the experimental power of *C. elegans* by taking multiple approaches to address the question at molecular, cellular and circuit levels. The findings are in general compelling. I have a few suggestions that will help complete the analysis.

1. Is *sri-36* expressed in any neurons other than ADF? The images in Figures 3g and 4d, 4f are too dark to see clearly. The authors should provide a more complete presentation on *Psri-36::GFP* expression.
2. The GCaMP6 responses in ADF are relatively small in amplitude – do the transgenic worms expressing GCaMP6 in ADF neuron display normal behavioral decisions in the diacetyl/glycerol paradigm? Also importantly, the authors need to provide their methods for calcium imaging, one of the major experimental approaches in this paper.
3. The authors need to document their source of *P. aeruginosa* PA14 strain.
4. In Figure 5d, it should be *ins-7* rescue, not *inx-7* rescue.

Reviewer #3

(Remarks to the Author)

The authors show that fasting and intestinal infection with *Pseudomonas aeruginosa* PA14 produce opposite decisions in a *C. elegans* multisensory conflict assay by acting through distinct gut–brain endocrine pathways that converge on the same sensory neurons. Fasting engages intestinal *AAK-2/DAF-16* to induce *INS-37*, whereas PA14 activates intestinal *PMK-2* to induce *INS-7*; both act on ADF neurons via *DAF-2* to upregulate the chemoreceptor *SRI-36*, but bias serotonin output toward approach or avoidance. The authors have an present impressive amount of work that defines a novel gut-brain communication pathway that dynamic sensory receptor tuning and divergent behavioral outcomes, offering a clear framework for how internal state reshapes sensory processing and decision-making. The depth of analysis is impressive including, identification chemoreceptor, Ca^{2+} imaging data and network analysis. The manuscript is clearly written and presents a compelling and conceptually novel gut–brain–behavior axis. I only have minor comments that could be clarified.

1. Other pathogens (*S. aureus* and EPEC) have no effect (Fig. 1b):

The mechanistic basis for this difference is unclear. Do these strains fail to colonize or persist after fasting, or do they not activate *PMK-2* signaling? Including colonization and persistence data after fasting, and/or *PMK-2* activation levels, would clarify this point. If feasible, testing additional pathogenic bacteria that colonize and persist after fasting and induce *PMK-2* immune signaling, and assessing whether they also reduce hunger-driven risk-taking, would further strengthen the argument.

2. Colonization vs pathogenicity:

Could it be that persistent gut bacteria could serve as internal “food-present” cues that suppress hunger responses? It would be informative to test a nonpathogenic strain that persists after fasting but does not activate *PMK-2*, to see if it fails to suppress escape.

3. PA14 under well-fed conditions (Supplementary Fig. 1b):

The authors report that PA14 does not alter decision-making in well-fed animals, but it is possible that PA14 still induces *PMK-2* immune signaling under these conditions. If so, the model might predict some reduction in risk-taking under a broader range of assay conditions, such as lower glycerol or higher diacetyl concentrations? For Fig. S1d and S1e, including well-fed controls for both OP50 and PA14 would strengthen interpretation. Fasting is known to increase diacetyl chemotaxis and fructose avoidance (ref. 6), yet these effects are not apparent here for glycerol avoidance. Testing a broader

range of diacetyl and glycerol concentrations could help reveal subtle effects.

5. 4 h vs 24 h infection (Supplementary Fig. 1g) and peptone supplementation (Fig. 1d):

Similar escape suppression seen after 4 h and 24 h PA14 exposure (S1g) is intriguing, as longer exposure might be expected to increase colonization and PMK-2 activation. Comparing colonization load and PMK-2 activation at both time points could clarify this. Likewise, while 0.35% peptone supplementation increases escape suppression, the effect is modest and may be influenced by outliers in the 0.25% condition.

6. Residual fasting effect without 5-HT or ADF:

Even with *tph-1* loss (Fig. 1h), ADF silencing (Fig. 1j), or *sri-36* deletion (Fig. 3f), fasting-induced escape remains higher than in well-fed animals. Including well-fed controls for both OP50 and PA14 in these experiments would help determine whether fasting still increases escape in these deficient backgrounds, suggesting the presence of parallel pathways.

7. *mod-5(n822)* mutant phenotype:

mod-5(n822) mutants, which have elevated extracellular 5-HT, show exaggerated fasting-induced escape (Fig. 1i). This seems inconsistent with the model, which predicts that high 5-HT should suppress escape in fasted worms, as chemogenetic activation of ADF reduces escape (Fig. 6h). Can the authors clarify

8. Exogenous 5-HT tests:

It could be informative to test exogenous 5-HT supplementation in WT and *tph-1* mutants under well-fed conditions, to see if low doses mimic fasting (increased escape) and high doses mimic PA14 infection (suppressed escape). Assessing effects on *sri-36* expression, ADF activity, and diacetyl sensitivity would connect pharmacological effects to the proposed pathway.

9. *sri-36* expression levels:

The “moderate vs high” SRI-36 expression model could be strengthened by expressing *sri-36* in ADF at moderate and high levels in *sri-36* mutants, then measuring ADF diacetyl sensitivity, 5-HT release, and risk-taking. This would test whether moderate expression reproduces fasting effects and high expression reproduces PA14 effects, strengthening the link between receptor expression and behavioral tuning.

10. ILP release mutants:

hid-1 and *egl-3* loss abolish ADF responses (Fig. 5a), consistent with the model that intestinal ILPs act via ADF DAF-2, *unc-31(e169)* mutants show no effect on ADF responses in any state (Fig. 2c). Does this mean that *unc-31* is not required for INS-37/7 release from the intestine?

Other minor

11. Supplementary Fig. 2 is not referenced in the main text.

12. line 295 The authors should weaken the statement: “These results demonstrate that hunger and *P. aeruginosa* infection induce intestinal INS-37 and INS-7 release, respectively” to indicate are consistent with.

13. Different bacterial strains were cultured on distinct media and under different incubation conditions (e.g., NGM vs. peptone-supplemented NGM, tryptic soy agar, modified NGM with tryptophan; varied incubation times and temperatures). It would be helpful if the authors could briefly explain the rationale for these differences—for example, whether they are required for optimal bacterial growth, virulence factor expression, or compatibility with the assay conditions.

14. For exogenous 5-HT stimulation, the authors applied serotonin directly to exposed neuronal somas via pressure ejection after incision. This is technically challenging. It would be useful to explain why this approach in more detail and why this was chosen instead of exposing 5-HT in intact animals during imaging. It would help readers understand the methodological rationale.

15. For Fig. S1f, fasting typically increases locomotion speed and roaming, which is not observed here? It would be helpful to explain this difference. For Fig. S1f, fasting is generally associated with increased locomotion speed, which is not observed under the current assay conditions. It would be helpful if the authors could explain why they chose this specific setup (30 s acclimation followed by 1 min recording at 15 fps) and whether these parameters might limit detection of fasting-induced locomotion changes.

16. Supplementary Fig. 2 is not referenced in the main text.

Reviewer #4

(Remarks to the Author)

In this submitted manuscript, Lei, Chen et al examine how *C. elegans* behavioral decisions are modulated by internal state. They use a classic paradigm where worms need to cross an aversive barrier to get to a chemoattractant Diacetyl. The

authors first show that fasted animals cross the barrier more frequently, whereas *P. aeruginosa*-infected animals that are fasted do not. They find that these behavioral changes are dependent on serotonin production by ADF neurons. ADF is shown to directly respond to diacetyl, scaling up its response in fasted animals and even further upon infection. The authors then identify the *sri-36* chemoreceptor as an ADF chemoreceptor that is upregulated upon fasting and even further upon infection. They show that the fasting effects work through AAK-2 and Insulin signaling, whereas the infection effects work through PMK/NSY/SEK signaling. Finally, the authors show that the fasting effects require the interneurons AIB/AUA/URX, whereas the infection effects require AIA/AIY. Serotonin receptors in these interneurons are identified.

Most of the analyses in this paper are well done. I appreciated the identification of SRI-36 as an induced chemoreceptor and the important role of ADF in this behavior. However, I have two main reservations about publication of this study in Nature Communications:

1) I worry that the final conceptual model in the paper doesn't quite make sense in the context of the field. First, the authors argue for a biphasic curve where ADF promotes escape when it releases a medium amount of serotonin, but not when it is inactive or releasing a lot of serotonin. It's just not clear how this could work. How would these subtly different levels of release result in vastly different behavioral responses? Showing how this works would require measuring dose-dependent serotonin responses in downstream interneurons much more carefully. Perhaps quantifying speed/turning and other parameters across these 3 internal states would help also. Moreover, the involvement of the interneurons doesn't seem congruent with a substantial amount of prior knowledge in the field. The authors argue that AIB promotes escape behavior, but all prior work suggests that AIB enhances reversals (which would lead to a naïve guess that it inhibits escape behavior). The authors also argue that AIA/AIY promote staying behavior, but all prior work suggests these neurons drive forward movement (which would lead to a naïve guess that they promote escape). Overall, I worry that this 'neural circuit mechanism' part of the paper is underdeveloped and may not be correctly worked out yet.

2) Regarding the first half of the paper – I find these results convincing and I appreciate the work that the authors put into this. But this is very well traveled ground. The idea that chemoreceptors are induced by fasting and infection has been established. The idea that Insulin signaling is involved in this process is also very well established. Relevant citations include: van der Linden et al 2007, Kyani-Rogers et al, 2022, Mclachlan et al, 2022, Wu et al, 2023, to name a few.

I also have a few other minor comments:

1. For the *sri-36* gene expression studies, it would be preferable to use a CRISPR knock-in reporter, which has become the standard state-of-the-art in the field for these types of studies.
2. Is *pmk/sek* signaling involved in the fasting effects? And is *aaak-2* involved in the infection effects? (or are these pathways totally separate and specific?)
3. ADF releases other neurotransmitters – are they involved in these behaviors?

Version 1:

Reviewer comments:

Reviewer #1

(Remarks to the Author)

The revised manuscript by Liu and colleagues does not address the principal concern raised in my initial review. The authors do not seem to grasp the seriousness of their lack of understanding of their own experimental system.

The authors state that their method is one that is widely accepted. In fact, the issue of whether dsRNA can spread from neurons has been examined by Jose and Hunter and colleagues. Specifically, as noted by Ravikumar et al. (NAR (2019) 47, 10059), "... dsRNA expressed in neurons can silence a target gene in somatic tissues such as the intestine, muscle, and hypodermic (refs 8, 17, 18) and in the germline (19)." Refs 8, 17, 18 are as follows:

8. Jose A.M., Smith J.J., Hunter C.P. Export of RNA silencing from *C. elegans* tissues does not require the RNA channel SID-1. Proc. Natl. Acad. Sci. U.S.A. 2009; 106:2283–2288.

17. Jose A.M., Garcia G.A., Hunter C.P. Two classes of silencing RNAs move between *C. elegans* tissues. Nat. Struct. Mol. Biol. 2011; 18:1184–1188.

18. Raman P., Zaghab S.M., Traver E.C., Jose A.M. The double-stranded RNA binding protein RDE-4 can act cell autonomously during feeding RNAi in *C. elegans*. Nucleic Acids Res. 2017; 45:8463–8473.

The authors state that neurons do not express SID-1. This is not likely the case based on the analysis of neuron-specific gene expression (CeNGEN). Jose et al. have also shown that SID-1 is not required for the export of dsRNA to other tissues (though it is required for the import).

Moreover, in their so-called "intestine-specific" RNAi experiments, the authors claim they are "confirming" their results by using a *sid-1* mutant background, but in fact because of the possibility of spreading, their initial experiments are not meaningful. The only tissue-specific RNAi experiments performed by the authors that use generally accepted methodology are limited to those using the VP303 strain in the analysis of intestine-specific insulins, as noted in my initial review. It may be helpful for the authors to review the rationale for the use of these strains (and other strains utilized for tissue-specific RNAi). To be clear, the issue is not "off-target effects" stated by the authors in their response, but rather, the targeted gene will be affected in a tissue and cell type distinct from what was intended.

The authors refer to "multiple orthogonal lines of evidence," but the only ones that relate to neuron-specific issues may be the use of cell-specific rescue, which is spotty and only done in limited instances, and in any case, does not establish necessity, only supports sufficiency. The key results derived from what the authors perceived were neuron-specific RNAi experiments are not supported by "multiple orthogonal lines of evidence."

The authors refer to their prior 2017 study that utilized the same methodology. I am not aware of other manuscripts using the same method, though the concerns stated here would still apply. Given the experimental concerns outlined here, the authors' 2017 manuscript also invites considerable skepticism.

Reviewer #2

(Remarks to the Author)

The authors fully addressed my questions in their revised manuscript.

Reviewer #3

(Remarks to the Author)

I think that the authors have adequately addressed the comments made by the reviewers in the revised version of the manuscript. Therefore, I have no further comments.

Reviewer #4

(Remarks to the Author)

The authors have addressed my concerns for the most part. I remain a bit skeptical of the overall model for the reasons stated in my original review, but if the authors are convinced then perhaps it makes sense to publish and let the field digest the results.

Reviewer #5

(Remarks to the Author)

Response to Reviewers' Comments

We thank the reviewers for their thorough evaluation of our manuscript and for their constructive feedback. We have carefully addressed each point, performed additional experiments, and made substantial changes to strengthen the study. Below, we provide point-by-point responses. For clarity, reviewer comments are reproduced in *italics*, followed by our responses, with reference to the relevant text, figure numbers, and references in the revised manuscript. All corresponding changes have been incorporated into the revised version and are highlighted in blue.

REVIEWER COMMENTS

Reviewer #1 (Remarks to the Author):

Lei et al. present a study that examines the role of gut-to-brain signaling in the simple animal C. elegans in modulating decision-making and behavioral responses to bacterial food cues and pathogenicity. The manuscript gets off to a strong start. The authors set up a nice assay that assesses the "risk-taking" behavior in crossing a noxious chemical barrier to access a food signal. The authors then examine how prior exposure to nutritious bacteria as well as three pathogenic bacteria, and also the effect of bacterial food removal, influence this decision making. The authors carry out a number of solid control experiments and confirm that fasting has a marked effect on subsequent behavioral responses. The authors find that exposure to pathogenic Pseudomonas aeruginosa reduces the animal's risk-taking food-seeking behavior. Genetic experiments with tph-1 suggest a role for TPH-1 and the ADF neurons in mediating this response. The authors then conduct calcium-imaging experiments of ADF responses to show how ADF activity is influenced during conditions of fasting and Pseudomonas exposure. The authors use a candidate expression approach to identify a putative receptor, SRI-36, that is strongly induced in the ADF neurons by fasting and even more so by Pseudomonas infection. The authors show that ADF responses are depending on AAK-2, DAF-16, and the NSY-1-SEK-1 p38 pathway. Up until this point, the manuscript is quite impressive with a combination of beautiful genetics and imaging.

Response: We thank the reviewer for the overall positive assessment of our work.

However, once the manuscript begins to address site-of-action issues, there are serious concerns with the underlying method being used. The authors claim to be carrying out "cell-specific RNAi." Methods are somewhat cursory, but it appears that

the primary method being utilized is to use cell- or tissue-specific promoters to express + and – strands of a gene of interest off of a plasmid that has been introduced into cells of interest by the generation of transgenic strains. What is not taken into account is that this method does not take into account the ability for dsRNA to spread from cell-to-cell. For this reason, cell-specific RNAi methods are generally performed in an RNAi-defective strain with cell-specific rescue of RNAi (e.g., rde-1). The VP303 strain used for the screen for intestine-specific insulins has these attributes. But none of the other ostensibly cell-specific RNAi experiments utilize this strain or have any indication of a cell-specific rescue of RNAi activity. Thus, one cannot conclude anything about cell necessity from these site-of-action experiments. (It appears that the author used this method in a prior 2020 Nature Communications article as well.)

Unless there is a trivial explanation that I am missing, this basic experimental oversight seriously undermines the confidence of this reviewer in the basic methodology and rigor of the manuscript. This is especially disappointing as the first part of the manuscript is remarkably impressive.

Response: We thank the reviewer for raising this important methodological concern. In our study, we employed a well-established tissue-specific RNAi strategy by co-injecting plasmids encoding complementary sense and antisense fragments of the target gene under cell-specific promoters. This approach has been widely used in *C. elegans* and is considered reliable because, except for a few phasmid and male ray neurons, most neurons are intrinsically resistant to systemic RNAi owing to the absence of the dsRNA transporter SID-1, which mediates inter-tissue RNAi spreading (Winston, W. M., Molodowitch, C. & Hunter, C. P., *Science*, 2002). Consistent with this, our previous work demonstrated that AVA-specific RNAi did not spread to A-type motor neurons, and vice versa (Liu et al., *Nat Commun*, 2017), validating the spatial restriction of this method.

Beyond RNAi, our conclusions are supported by multiple orthogonal lines of evidence, including analyses of loss-of-function mutants and cell-specific rescues (*sri-36*, *aak-2*, *ins-7*, *ser-4*, *lgc-50*, and *mod-1*), as well as cell-specific expression of the constitutively nuclear-localized DAF-16 (DAF-16a^{AM}). These independent approaches consistently converge on the same mechanistic pathway, underscoring the robustness of our conclusions.

Nevertheless, we recognize the reviewer's point that potential off-target effects in

IECs cannot be completely excluded. To rigorously address this concern, we repeated the key intestine-specific RNAi experiments (*aak-2*, *daf-16*, *nsy-1*, *sek-1*, *pmk-2*, *hid-1*, *ins-37*, and *ins-7*) in the systemic RNAi-defective *sid-1(qt9)* mutant background, in which dsRNA spreading is abolished (Winston, W. M., Molodowitch, C. & Hunter, C. P., *Science*, 2002). The results (revised Fig. 4a-g and Fig. 5a,c,d,f,g) are consistent with our original findings, confirming that the observed effects reflect intestine-specific knockdown. These findings are now described in the Results and clarified in the revised Methods as follows: “To confirm tissue specificity and exclude systemic RNAi effects, intestine-specific RNAi experiments (*aak-2*, *daf-16*, *nsy-1*, *sek-1*, *pmk-2*, *hid-1*, *ins-37*, and *ins-7*) were repeated in the systemic RNAi-defective *sid-1(qt9)* mutant background, in which dsRNA spreading is abolished.”

Reviewer #2 (Remarks to the Author):

“Intestinal pathogens override hunger-driven decision-making via immunoregulation of central serotonin signaling in C. elegans”

Serotonin is known to react to various environmental stresses and internal states to modulate neural functions and generate behavioral responses. Some of these behavioral responses are orthogonal when the conditions are presented alone to the animals. How serotonin signaling regulates neural activity to produce coherent and balanced behavioral decisions under conditions demanding conflicting outputs from the nervous system is a fundamentally important but challenging question. In the above manuscript, Lei and Chen et al. set out to address this issue.

*The authors employed a risk-taking decision paradigm in C. elegans where the worms' attraction to an attractive odorant diacetyl (sensed by AWA sensory neuron) is hindered by a hyperosmotic glycerol barrier (sensed by ASH sensory neuron). They characterized how serotonin signaling modulated this decision-making process under two well-controlled conditions. While starvation promotes risk-taking as previously shown, the authors found that feeding on a pathogenic bacteria *Pseudomonas aeruginosa* PA14 suppressed the effect of starvation and inhibited the decision to cross the hyperosmotic barrier.*

*Using this assay together with cell-specific manipulation of serotonin production and neural activity, the authors showed that starvation and *P. aeruginosa* infection converged on the serotonin signal produced by ADF neurons to regulate behavior*

decision on diacetyl/glycerol response, and that starvation and infection both induced ADF response to diacetyl. Using previously published expression profiles, RNAi, and calcium imaging, CRISPR and cell-specific rescue, the authors further showed that starvation and *P. aeruginosa* infection upregulated the expression of a GPCR *sri-36* in ADF and that *sri-36* mediated diacetyl responses in fasted and infected worm in ADF neurons and in behavioral decision. The authors further identified two intestinal pathways, the *aak-2/daf-16* pathway and *pmk-2/daf-16* pathway, that upregulated *sri-36* under fasting and infection conditions. These pathways upregulated expression of two insulin-like peptides, *ins-37* and *ins-7*, in fasted and infected worms to modulate ADF expression of *sri-36* and the behavioral decision between diacetyl/glycerol. Furthermore, the authors mapped the downstream neural circuit and serotonin receptors that mediated starvation and infection-regulated diacetyl response in ADF and behavioral decision, and demonstrated that a stronger release of serotonin from ADF in infected worms compared with starved worms suppressed the risk-taking effect of starvation. Based on these findings, they delineate a molecular strategy in a neuronal circuit that integrates food availability and infection to produce a balanced behavior decision between safety-seeking or risk-taking. The authors fully leverage the experimental power of *C. elegans* by taking multiple approaches to address the question at molecular, cellular and circuit levels. The findings are in general compelling. I have a few suggestions that will help complete the analysis.

Response: We thank the reviewer for the overall positive assessment of our work.

1. *Is sri-36 expressed in any neurons other than ADF? The images in Figures 3g and 4d, 4f are too dark to see clearly. The authors should provide a more complete presentation on Psri-36::GFP expression.*

Response: We thank the reviewer for raising this important point. We have improved *Psri-36::GFP* images and now included a whole-body view. The data show that *sri-36* expression is restricted to ADF neurons during *P. aeruginosa* infection, with no detectable expression in other neurons (new Supplementary Fig. 3b). To further validate this observation, we generated a CRISPR knock-in *sri-36::mNeonGreen* reporter with mNeonGreen inserted at the C-terminus of the endogenous locus. This knock-in reporter corroborated the *Psri-36::GFP* expression pattern (new Supplementary Fig. 3c). These findings are now incorporated into the Results.

2. *The GCaMP6 responses in ADF are relatively small in amplitude – do the transgenic worms expressing GCaMP6 in ADF neuron display normal behavioral*

decisions in the diacetyl/glycerol paradigm? Also importantly, the authors need to provide their methods for calcium imaging, one of the major experimental approaches in this paper.

Response: We agree that it is important to confirm that GCaMP6 expression in ADF does not interfere with behavior. We tested the transgenic worms expressing GCaMP6 in ADF neuron (*xyhEx446[Psrh-142::HisCl1::SL2::GCaMP6s, Punc-122::GFP]*) and found that they exhibited normal escape behavior compared to wild type, indicating that GCaMP6 expression does not affect decision-making (new Supplementary Fig. 2e). We have also expanded the Methods to include a detailed description of the calcium imaging protocol (revised Methods, “Imaging”).

3. The authors need to document their source of P. aeruginosa PA14 strain.

Response: We thank the reviewer for this suggestion. The bacterial strains used in this study are now listed in Table 1 and were obtained from the following sources: *E. coli* OP50 from the Caenorhabditis Genetics Center (CGC); *P. aeruginosa* PA14 and *S. aureus* NCTC8325 from Dr. Jianke Gong (Huazhong University of Science and Technology); PA14-dsRed from Dr. Bin Qi (Yunnan University); EPEC and *E. coli* HT115 from Dr. Anbing Shi (Huazhong University of Science and Technology); and PA14-*gacA*(-) from Taihong Wu (Hunan University).

4. In Figure 5d, it should be ins-7 rescue, not inx-7 rescue.

Response: Corrected, thanks.

Reviewer #3 (Remarks to the Author):

The authors show that fasting and intestinal infection with Pseudomonas aeruginosa PA14 produce opposite decisions in a C. elegans multisensory conflict assay by acting through distinct gut–brain endocrine pathways that converge on the same sensory neurons. Fasting engages intestinal AAK-2/DAF-16 to induce INS-37, whereas PA14 activates intestinal PMK-2 to induce INS-7; both act on ADF neurons via DAF-2 to upregulate the chemoreceptor SRI-36, but bias serotonin output toward approach or avoidance. The authors have an present impressive amount of work that defines a novel gut-brain communication pathway that dynamic sensory receptor tuning and divergent behavioral outcomes, offering a clear framework for how internal state reshapes sensory processing and decision-making. The depth of

analysis is impressive including, identification chemoreceptor, Ca²⁺ imaging data and network analysis. The manuscript is clearly written and presents a compelling and conceptually novel gut–brain–behavior axis. I only have minor comments that could be clarified.

Response: We thank the reviewer for the overall positive assessment of our work.

1. Other pathogens (S. aureus and EPEC) have no effect (Fig. 1b):

The mechanistic basis for this difference is unclear. Do these strains fail to colonize or persist after fasting, or do they not activate PMK-2 signaling? Including colonization and persistence data after fasting, and/or PMK-2 activation levels, would clarify this point. If feasible, testing additional pathogenic bacteria that colonize and persist after fasting and induce PMK-2 immune signaling, and assessing whether they also reduce hunger-driven risk-taking, would further strengthen the argument.

Response: We thank the reviewer for this insightful comment. We have now performed colonization assays for *S. aureus* and EPEC after fasting, showing that *S. aureus* colonized the intestine whereas EPEC did not, but neither pathogen induced *Pins-7::GFP* expression (new Supplementary Fig. 5e,f). These results are consistent with their lack of behavioral effect, indicating that activation of the intestinal PMK-2 immune signaling pathway is required to suppress hunger-driven risk-taking. These findings are now incorporated into the Results and further discussed in the Discussion. We agree that testing additional pathogens will be valuable to generalize this mechanism and have emphasized this point in the revised Discussion as follows: “Future studies will be important to test additional pathogens that both colonize and activate PMK-2 signaling, to determine whether this mechanism for overriding hunger-driven decision-making is broadly conserved across diverse microbial infections.”

2. Colonization vs pathogenicity:

Could it be that persistent gut bacteria could serve as internal “food-present” cues that suppress hunger responses? It would be informative to test a nonpathogenic strain that persists after fasting but does not activate PMK-2, to see if it fails to suppress escape.

Response: We thank the reviewer for this thoughtful comment. However, our results argue against this possibility and instead support that suppression of hunger responses requires immune activation by *P. aeruginosa*:

1. Pathogen specificity: Among the three pathogens tested, only *P. aeruginosa* suppressed hunger-driven escape, whereas *S. aureus* colonized the intestine but had no effect (Fig. 1b and new Supplementary Fig. 5e). Knockdown of *nol-6*, which reduces *P. aeruginosa* colonization, attenuated suppression (Fig. 1e,f). These results indicate that colonization is necessary but not sufficient.

2. Virulence requirement: Suppression required live and virulent *P. aeruginosa*. Neither supernatant, odors, heat-killed cells, nor the nonpathogenic PA14-*gacA*(-) mutant suppressed escape (Fig. 1c). Conversely, enhancing virulence with 0.35% peptone further strengthened suppression (Fig. 1d). Thus, virulence rather than persistence alone is critical.

3. Immune signaling dependence: Genetic analyses showed that suppression depends on intestinal PMK-2/p38 MAPK signaling (Fig. 4), which is activated by *P. aeruginosa* but not by *S. aureus* (new Supplementary Fig. 5f), establishing a mechanistic requirement for immune activation.

4. Neural and sensory signatures: Infection produced distinct physiological and neural changes not observed with simple feeding. Specifically, SRI-36 expression in ADF, diacetyl-evoked calcium responses in ADF and AWA, and glycerol-evoked calcium responses in ASH all differed markedly between well-fed and *P. aeruginosa*-infected worms, arguing against infection simply mimicking food presence.

5. Nonpathogenic strain test: We examined the nonpathogenic *P. aeruginosa* mutant strain PA14-*gacA*(-) and found that it colonized the intestine nearly 16-fold more than *E. coli* after fasting (CFUs: 555 ± 87.63 vs. 35.33 ± 10.64) (new Supplementary Fig. 5e). However, PA14-*gacA*(-) did not induce *Pins-7::GFP* expression (new Supplementary Fig. 5f) and failed to suppress hunger-driven escape (Fig. 1c), in contrast to wild-type *P. aeruginosa*.

3. PA14 under well-fed conditions (Supplementary Fig. 1b):

The authors report that PA14 does not alter decision-making in well-fed animals, but it is possible that PA14 still induces PMK-2 immune signaling under these conditions. If so, the model might predict some reduction in risk-taking under a broader range of assay conditions, such as lower glycerol or higher diacetyl concentrations? For Fig. S1d and S1e, including well-fed controls for both OP50 and PA14 would strengthen interpretation. Fasting is known to increase diacetyl chemotaxis and fructose avoidance (ref. 6), yet these effects are not apparent here for glycerol avoidance.

Testing a broader range of diacetyl and glycerol concentrations could help reveal subtle effects.

Response: We thank the reviewer for this insightful comment. As expected, our new experiments showed that *P. aeruginosa* induced *Pins-7::GFP* expression in well-fed worms (new Supplementary Fig. 5g), indicating activation of PMK-2 signaling. Consistently, *P. aeruginosa* also induced *Psri-36::GFP* expression, but the signal was very weak (new Supplementary Fig. 5h). To address the reviewer's suggestion regarding behavioral effects:

1. We added well-fed controls for both *E. coli* and *P. aeruginosa* in the diacetyl and glycerol assays (revised Supplementary Fig. 1d,e). As expected, fasting increased diacetyl chemotaxis (revised Supplementary Fig. 1d). For glycerol avoidance in the absence of diacetyl, fasting increased escape at 2 M glycerol, but not at 1 M (likely too weak) or 3 M (likely because the motivation to escape was outweighed by the inhibitory effect of this high glycerol concentration on locomotion) (revised Supplementary Fig. 1e). Nevertheless, there was no significant difference between *E. coli*- and *P. aeruginosa*-fed worms under either well-fed or fasted conditions in both unisensory assays (revised Supplementary Fig. 1d,e). These findings are consistent with previous results (Ghosh, D. D. et al. *Neuron*, 2016) and with our data showing that fasting enhanced AWA diacetyl responses and reduced ASH glycerol responses, whereas *P. aeruginosa* infection did not further alter their responses (Supplementary Fig. 2b,c). Together, these results support our conclusion that *P. aeruginosa* does not affect unisensory decision-making.

2. As suggested, we also tested additional glycerol concentrations (1 M and 2 M) paired with 1:1000 diacetyl in well-fed worms. Consistent with the results at 3 M glycerol (Supplementary Fig. 1b), *P. aeruginosa* infection did not affect escape under these conditions (new Supplementary Fig. 5i), suggesting that immune activation has only subtle effects on risk-taking in the absence of fasting.

Together, these results suggest that while *P. aeruginosa* activates intestinal immune signaling in both well-fed and fasted worms, its behavioral suppression is most pronounced under fasting, when hunger-driven risk-taking is elevated. These findings are now incorporated into the Results.

5. *4 h vs 24 h infection (Supplementary Fig. 1g) and peptone supplementation (Fig. 1d):*

Similar escape suppression seen after 4 h and 24 h PA14 exposure (S1g) is intriguing, as longer exposure might be expected to increase colonization and PMK-2 activation. Comparing colonization load and PMK-2 activation at both time points could clarify this. Likewise, while 0.35% peptone supplementation increases escape suppression, the effect is modest and may be influenced by outliers in the 0.25% condition.

Response: We thank the reviewer for this insightful comment. To address these points, we quantified both intestinal PA14-dsRed colonization and *Pins-7::GFP* expression after 4 h and 24 h of infection (new Supplementary Fig. 1i and new Supplementary Fig. 5a), and under 0.25% vs. 0.35% peptone supplementation (new Supplementary Fig. 1h and new Supplementary Fig. 5b).

1. 4 h vs. 24 h infection: PA14-dsRed colonization and *Pins-7::GFP* expression did not show significant difference, consistent with the similar escape behavior. These results suggest that infection and immune activation reach a saturating level by 4 h.

2. 0.25% vs. 0.35% peptone: Both PA14-dsRed colonization and *Pins-7::GFP* expression were higher at 0.35% peptone, consistent with the stronger escape suppression. These results indicate that enhanced virulence promotes colonization and immune activation, but the behavioral response is limited by ceiling effects.

These findings have been incorporated into the Results.

6. Residual fasting effect without 5-HT or ADF:

*Even with *tph-1* loss (Fig. 1h), ADF silencing (Fig. 1j), or *sri-36* deletion (Fig. 3f), fasting-induced escape remains higher than in well-fed animals. Including well-fed controls for both *OP50* and *PA14* in these experiments would help determine whether fasting still increases escape in these deficient backgrounds, suggesting the presence of parallel pathways.*

Response: We thank the reviewer for this insightful comment. We have now included well-fed controls for *E. coli* and *P. aeruginosa* in these deficient backgrounds (new Supplementary Fig. 8). These data show that fasting still increased escape even when 5-HT synthesis, ADF activity, or *sri-36* was disrupted, indicating the involvement of 5-HT-independent, parallel mechanisms. We have expanded the Discussion to note this point as follows: "Notably, fasting still increased escape even when 5-HT synthesis, ADF activity, or *sri-36* was disrupted, suggesting the presence of parallel, 5-HT-independent pathways. These may include other

neuromodulators such as tyramine, dopamine, and additional signals that contribute to hunger-driven risk-taking. Future work will be needed to delineate these parallel mechanisms and determine how they interact with the 5-HT axis to shape adaptive decision-making.”

7. mod-5(n822) mutant phenotype:

mod-5(n822) mutants, which have elevated extracellular 5-HT, show exaggerated fasting-induced escape (Fig. 1i). This seems inconsistent with the model, which predicts that high 5-HT should suppress escape in fasted worms, as chemogenetic activation of ADF reduces escape (Fig. 6h). Can the authors clarify

Response: We agree that the *mod-5(n822)* phenotype may appear paradoxical at first glance. We propose that this difference reflects the spatial and temporal dynamics of 5-HT signaling. In *mod-5(n822)* mutants, defective reuptake elevates basal extracellular 5-HT levels, producing a chronic and diffuse increase that likely potentiates hunger-driven escape by enhancing overall 5-HT tone. In contrast, chemogenetic activation of ADF induces a strong, acute, and spatially restricted 5-HT release at ADF synapses, which suppresses risk-taking. Thus, the *mod-5(n822)* phenotype is consistent with our model: moderate increases in 5-HT release promotes risk-taking, whereas excessive ADF-specific release suppresses it.

8. Exogenous 5-HT tests:

*It could be informative to test exogenous 5-HT supplementation in WT and *tph-1* mutants under well-fed conditions, to see if low doses mimic fasting (increased escape) and high doses mimic PA14 infection (suppressed escape). Assessing effects on *sri-36* expression, ADF activity, and diacetyl sensitivity would connect pharmacological effects to the proposed pathway.*

Response: We thank the reviewer for this insightful suggestion. We did not pursue exogenous 5-HT supplementation in this study for two reasons. First, exogenous 5-HT broadly activates serotonergic targets throughout the animal, whereas our model focuses on state-dependent modulation of ADF-derived 5-HT and its circuit-specific effects. Second, exogenous 5-HT can produce pleiotropic and dose-dependent outcomes that are difficult to interpret, as its effects vary with concentration, timing, and site of application. Instead, we directly tested our model using targeted genetic and physiological approaches by monitoring *sri-36* expression, ADF activity, and ADF 5-HT release (synapto-pHluorin imaging). These complementary strategies allowed us to causally link intestinal signaling, ADF chemoreceptor expression, and behavior.

9. *sri-36* expression levels:

The “moderate vs high” SRI-36 expression model could be strengthened by expressing *sri-36* in ADF at moderate and high levels in *sri-36* mutants, then measuring ADF diacetyl sensitivity, 5-HT release, and risk-taking. This would test whether moderate expression reproduces fasting effects and high expression reproduces PA14 effects, strengthening the link between receptor expression and behavioral tuning.

Response: We thank the reviewer for this thoughtful suggestion. We previously injected 25 ng/μl of the *Psrh-142::sri-36* plasmid and observed rescue of both ADF diacetyl responses and escape in fasted and infected worms (Fig. 3e,f). However, the exact expression levels could not be determined in the absence of a fluorescent marker. To directly test the effect of expression level, we expressed SRI-36 at two different levels in *sri-36(plc921)* mutants by injecting 5 ng/μl (“low” expression) or 75 ng/μl (“high” expression) of the *Psrh-142::sri-36::mStrawberry* plasmid. mCherry fluorescence intensity confirmed lower SRI-36 expression in ADF at 5 ng/μl compared to 75 ng/μl (Supplementary Fig. 3d). In fasted worms, low-level expression restored moderate ADF diacetyl responses, 5-HT release, and escape, resembling fasted controls. In contrast, high-level expression enhanced ADF diacetyl responses and 5-HT release, mimicking the suppression of escape observed during *P. aeruginosa* infection (revised Fig. 3e,f and Fig. 6g). These results directly support our model, demonstrating that SRI-36 expression levels tune ADF sensitivity, 5-HT output, and behavioral choice. The corresponding findings have been incorporated into the Results.

10. ILP release mutants:

hid-1 and *egl-3* loss abolish ADF responses (Fig. 5a), consistent with the model that intestinal ILPs act via ADF DAF-2, *unc-31(e169)* mutants show no effect on ADF responses in any state (Fig. 2c). Does this mean that *unc-31* is not required for INS-37/7 release from the intestine?

Response: We thank the reviewer for this important point. Based on our results (Fig. 2c), UNC-31 is not required for fasting-induced INS-37 release or *P. aeruginosa*-induced INS-7 release from IECs under our experimental conditions. This is consistent with published reports showing that UNC-31 is not expressed in the intestine (Speese, S. et al., *J Neurosci*, 2007; Cornell, R., Cao, W., Liu, J. & Pocock, R., *J Neurosci*, 2022).

Other minor

11. *Supplementary Fig. 2 is not referenced in the main text.*

Response: We thank the reviewer for pointing this out. This figure was inadvertently included from an earlier draft and is not part of the current study. We have removed it from the revised submission.

12. *line 295 The authors should weaken the statement : “These results demonstrate that hunger and P. aeruginosa infection induce intestinal INS-37 and INS-7 release, respectively” to indicate are consistent with.*

Response: We thank the reviewer for this helpful suggestion. We have revised the statement to: “These results are consistent with a model in which hunger and *P. aeruginosa* infection induce intestinal INS-37 and INS-7 release, respectively,”

13. *Different bacterial strains were cultured on distinct media and under different incubation conditions (e.g., NGM vs. peptone-supplemented NGM, tryptic soy agar, modified NGM with tryptophan; varied incubation times and temperatures). It would be helpful if the authors could briefly explain the rationale for these differences—for example, whether they are required for optimal bacterial growth, virulence factor expression, or compatibility with the assay conditions.*

Response: We thank the reviewer for this insightful comment. Distinct culture conditions were selected to ensure optimal growth, virulence factor expression, and compatibility with *C. elegans* assays (Wu, T. et al., *Nature*, 2023; Anyanful, A., Easley, K. A., Benian, G. M. & Kalman, *Cell Host Microbe*, 2009). Specifically:

1. *P. aeruginosa* was plated on NGM + 0.25% peptone, which supports robust virulence factor production while maintaining animal viability for behavioral assays. 0.35% peptone was used to further enhance virulence.

2. *S. aureus* was cultured on tryptic soy agar with nalidixic acid, which both optimizes growth of this Gram-positive pathogen and prevents contamination.

3. EPEC was grown on modified NGM containing tryptophan, a condition previously reported to enhance its virulence toward *C. elegans*.

We have added these explanations into the Methods (“Bacterial strains and culture”).

14. *For exogenous 5-HT stimulation, the authors applied serotonin directly to exposed neuronal somas via pressure ejection after incision. This is technically challenging. It*

would be useful to explain why this approach in more detail and why this was chosen instead of exposing 5-HT in intact animals during imaging. It would help readers understand the methodological rationale.

Response: We thank the reviewer for this thoughtful comment. The targeted interneurons (AIA, AIY, AIB, and AUA) are located deep within the head and are not directly accessible to exogenously applied 5-HT in intact animals. Therefore, exposing intact worms to 5-HT would not allow precise or reproducible stimulation of these neurons. To overcome this limitation, we used an established approach (Zhan, X. et al., *Nat Commun*, 2023), in which immobilized animals are incised along the glued region to expose neuron somas. 5-HT was then directly applied onto the soma by pressure ejection, enabling spatially precise, dose-controlled, and reproducible stimulation. We have clarified this rationale in the revised Methods (“Imaging”).

15. For Fig. S1f, fasting typically increases locomotion speed and roaming, which is not observed here? It would be helpful to explain this difference. For Fig. S1f, fasting is generally associated with increased locomotion speed, which is not observed under the current assay conditions. It would be helpful if the authors could explain why they chose this specific setup (30 s acclimation followed by 1 min recording at 15 fps) and whether these parameters might limit detection of fasting-induced locomotion changes.

Response: We thank the reviewer for raising this important point. Although fasting is often thought to increase locomotion or roaming, published evidence indicates that this effect is context-dependent. For example, Sawin et al. (*Neuron*, 2000) reported no significant difference in locomotory rate (body bends) between 30 min-fasted and well-fed animals, consistent with our results obtained using the Track-A-Worm automated tracking system. Regarding our recording setup, we used a 30 s acclimation and 1 min recording at 15 fps to minimize environmental variability and ensure reproducibility across groups. This protocol provides robust sampling of basal locomotion aligned with our behavioral paradigms. While longer recordings might reveal subtle differences, both our data and Sawin et al. suggest that fasting does not markedly alter basal locomotion under these conditions.

16. Supplementary Fig. 2 is not referenced in the main text.

Response: We thank the reviewer for pointing this out. As noted in our response to comment 11, this figure was inadvertently included from an earlier draft and has been

removed from the revised manuscript.

Reviewer #4 (Remarks to the Author):

In this submitted manuscript, Lei, Chen et al examine how C. elegans behavioral decisions are modulated by internal state. They use a classic paradigm where worms need to cross an aversive barrier to get to a chemoattractant Diacetyl. The authors first show that fasted animals cross the barrier more frequently, whereas P. aeruginosa-infected animals that are fasted do not. They find that these behavioral changes are dependent on serotonin production by ADF neurons. ADF is shown to directly respond to diacetyl, scaling up its response in fasted animals and even further upon infection. The authors then identify the sri-36 chemoreceptor as an ADF chemoreceptor that is upregulated upon fasting and even further upon infection. They show that the fasting effects work through AAK-2 and Insulin signaling, whereas the infection effects work through PMK/NSY/SEK signaling. Finally, the authors show that the fasting effects require the interneurons AIB/AUA/URX, whereas the infection effects require AIA/AIY. Serotonin receptors in these interneurons are identified.

Most of the analyses in this paper are well done. I appreciated the identification of SRI-36 as an induced chemoreceptor and the important role of ADF in this behavior. However, I have two main reservations about publication of this study in Nature Communications:

Response: We thank the reviewer for the overall positive assessment of our work.

1) I worry that the final conceptual model in the paper doesn't quite make sense in the context of the field. First, the authors argue for a biphasic curve where ADF promotes escape when it releases a medium amount of serotonin, but not when it is inactive or releasing a lot of serotonin. It's just not clear how this could work. How would these subtly different levels of release result in vastly different behavioral responses? Showing how this works would require measuring dose-dependent serotonin responses in downstream interneurons much more carefully. Perhaps quantifying speed/turning and other parameters across these 3 internal states would help also. Moreover, the involvement of the interneurons doesn't seem congruent with a substantial amount of prior knowledge in the field. The authors argue that AIB promotes escape behavior, but all prior work suggests that AIB enhances reversals (which would lead to a naïve guess that it inhibits escape behavior). The authors also

argue that AIA/AIY promote staying behavior, but all prior work suggests these neurons drive forward movement (which would lead to a naïve guess that they promote escape). Overall, I worry that this ‘neural circuit mechanism’ part of the paper is underdeveloped and may not be correctly worked out yet.

Response: We thank the reviewer for this perceptive and constructive critique. Our data support a model in which moderate ADF activation (as during fasting) promotes risk-taking, whereas strong ADF activation (as during infection) suppresses it. We interpret this as a level-dependent mechanism: moderate 5-HT release preferentially engages the ADF-AIB/AUA pathway, while excessive release recruits the ADF-AIA/AIY pathway, producing opposing behavioral outcomes. Mechanistically, distinct 5-HT receptors and target neurons likely have different activation thresholds and functional responses, allowing graded 5-HT release to differentially recruit downstream circuits. This logic is consistent with whole-brain analyses showing that 5-HT modulation is highly site- and receptor-specific, with different 5-HT concentrations and release patterns engaging distinct network modules (Dag, U. et al., *Cell*, 2023; Flavell, Steven W. et al., *Cell*, 2013). We have clarified this conceptual framework in the Discussion and now noted that “Future work should quantify 5-HT dose-response effects on downstream interneurons (e.g., changes in speed or turning dynamics across internal states) to further refine this circuit model.”

Regarding interneuron assignments, we agree with the reviewer’s summary of prior literature and clarify that our model aligns with established neuronal functions (Metaxakis, A., Petratos, D. & Tavernarakis, N., *Open Biol*, 2018; Ezcurra, M. et al., *J Neurosci*, 2016):

1. AIB promotes avoidance; thus, its inhibition by moderate ADF activity reduces avoidance, facilitating risk-taking.
2. AUA suppresses avoidance and biases animals toward approach; its activation by moderate ADF activity promotes escape across the barrier.
3. AIA and AIY promote attraction and forward locomotion; under infection, strong ADF activation suppresses AIA/AIY, thereby reducing food approach.

Together, these results suggest that ADF flexibly recruits distinct interneuron modules to implement opposing behavioral strategies according to metabolic and immune states. Our imaging, chemogenetic, and behavioral data consistently support this framework.

2) Regarding the first half of the paper – I find these results convincing and I appreciate the work that the authors put into this. But this is very well traveled ground. The idea that chemoreceptors are induced by fasting and infection has been established. The idea that Insulin signaling is involved in this process is also very well established. Relevant citations include: van der Linden et al 2007, Kyani-Rogers et al, 2022, McLachlan et al, 2022, Wu et al, 2023, to name a few.

Response: We thank the reviewer for acknowledging the strength of the study and for highlighting relevant prior work. We agree that chemoreceptor regulation by internal states and the involvement of insulin signaling have been elegantly established in earlier studies (van der Linden, A. M., Nolan, K. M. & Sengupta, P., *EMBO J*, 2007; Kyani-Rogers, T. et al., *Genetics*, 2022; McLachlan, I. G. et al., *eLife*, 2022; Wu, T. et al., *Nature*, 2023). These studies showed that fasting, infection, or developmental history can modulate chemoreceptor expression through pathways involving DAF-16/FOXO, MEF2, or ZIP-5, thereby influencing sensory and behavioral responses. Our work substantially extends these concepts in several key ways:

1. Identification of SRI-36 as a diacetyl receptor or essential subunit: We demonstrate that SRI-36 is both necessary and sufficient to tune ADF diacetyl sensitivity and risk-taking behavior. To our knowledge, SRI-36 is the first chemoreceptor shown to be induced by both metabolic (fasting) and immune (pathogen) signaling in *C. elegans*.
2. Integration of immune and metabolic signaling via a FOXO-to-FOXO axis: We delineate how two distinct intestinal pathways, AAK-2-DAF-16 during fasting and PMK-2-DAF-16 during infection, converge on ADF, representing a previously unrecognized form of immune-metabolic integration for gut-to-brain communication.
3. Linking receptor tuning to circuit and behavioral outcomes: Beyond describing chemoreceptor regulation, we establish the causal pathway from intestinal signaling, through SRI-36 induction and ADF 5-HT release, to downstream interneurons that implement opposing behavioral strategies. This circuit-level framework connects intestinal signaling to defined neuronal pathways, substantially extending prior observations.
4. State-specific signaling dependence: Our findings reveal a gut-to-brain mechanism whereby hunger promotes risk-taking while infection suppresses it, both by modulating central 5-HT signaling through distinct intestinal pathways. This provides

mechanistic insight into how internal states reshape decision-making.

Together, our findings build on and go beyond previous work. We have cited these relevant studies in the revised Discussion.

I also have a few other minor comments:

1. For the sri-36 gene expression studies, it would be preferable to use a CRISPR knock-in reporter, which has become the standard state-of-the-art in the field for these types of studies.

Response: We thank the reviewer for this valuable suggestion. As suggested, we generated a CRISPR knock-in *sri-36::mNeonGreen* reporter, with mNeonGreen inserted at the C-terminus of the endogenous locus. This reporter confirmed that SRI-36 expression is specifically induced in ADF neurons upon fasting, further enhanced by *P. aeruginosa* infection, and localized to the sensory cilia (new Supplementary Fig. 3c). These results validate our original conclusions and support the role of SRI-36 in detecting environmental odors. The new data have been incorporated into the Results.

2. Is pmk/sek signaling involved in the fasting effects? And is aak-2 involved in the infection effects? (or are these pathways totally separate and specific?)

Response: We thank the reviewer for this insightful question. To address it, we performed reciprocal reporter assays. We found that fasting-induced *Pins-37::GFP* expression was not altered by intestinal *pmk-2* RNAi (new Supplementary Fig. 5c). In contrast, *P. aeruginosa* infection-induced *Pins-7::GFP* expression was further enhanced in *aak-2* RNAi worms after 1 h and 2 h of infection, but not at 4 h, suggesting a transient peak of infection-induced *ins-7* expression (new Supplementary Fig. 5d). These results indicate that *pmk-2* signaling is not required for fasting-induced *ins-37* expression, whereas *aak-2* normally acts to dampen *ins-7* induction during infection. Thus, while *aak-2* and *pmk-2* serve as primary mediators of fasting- and infection-induced responses, respectively, our findings also reveal limited cross-talk between these pathways. These new data have been incorporated into the Results.

3. ADF releases other neurotransmitters – are they involved in these behaviors?

Response: We thank the reviewer for this insightful question. Based on current knowledge, ADF is also cholinergic (Pereira, L. et al., *eLife*, 2015). To test whether

acetylcholine (ACh) contributes, we performed ADF-specific *unc-17* RNAi to reduce vesicular ACh release. This manipulation did not significantly affect escape in either the fasted or infected state (new Supplementary Fig. 1j). These findings indicate that ACh release from ADF is not required for the state-dependent regulation of decision-making, although it may contribute to other ADF-mediated functions. These new data have been incorporated into the Results.

Response to Reviewers' Comments

We thank the reviewers for their careful and constructive comments, which have substantially improved our manuscript. We are pleased that the revisions have addressed the concerns raised. Below, we provide point-by-point responses. For clarity, reviewer comments are reproduced in *italics*, followed by our responses, with reference to the relevant text in the revised manuscript. All changes have been incorporated into the revised version and are highlighted in blue.

REVIEWER COMMENTS

Reviewer #1 (Remarks to the Author):

The revised manuscript by Liu and colleagues does not address the principal concern raised in my initial review. The authors do not seem to grasp the seriousness of their lack of understanding of their own experimental system.

The authors state that their method is one that is widely accepted. In fact, the issue of whether dsRNA can spread from neurons has been examined by Jose and Hunter and colleagues. Specifically, as noted by Ravikumar et al. (NAR (2019) 47, 10059), "... dsRNA expressed in neurons can silence a target gene in somatic tissues such as the intestine, muscle, and hypodermic (refs 8, 17, 18) and in the germline (19)." Refs 8, 17, 18 are as follows:

8. Jose A.M., Smith J.J., Hunter C.P. Export of RNA silencing from C. elegans tissues does not require the RNA channel SID-1. Proc. Natl. Acad. Sci. U.S.A. 2009; 106:2283–2288.

17. Jose A.M., Garcia G.A., Hunter C.P. Two classes of silencing RNAs move between C. elegans tissues. Nat. Struct. Mol. Biol. 2011; 18:1184–1188.

18. Raman P., Zaghab S.M., Traver E.C., Jose A.M. The double-stranded RNA binding protein RDE-4 can act cell autonomously during feeding RNAi in C. elegans. Nucleic Acids Res. 2017; 45:8463–8473.

The authors state that neurons do not express SID-1. This is not likely the case based on the analysis of neuron-specific gene expression (CeNGEN). Jose et al. have also shown that SID-1 is not required for the export of dsRNA to other tissues (though it is required for the import).

Moreover, in their so-called "intestine-specific" RNAi experiments, the authors claim they are "confirming" their results by using a sid-1 mutant background, but in fact because of the possibility of spreading, their initial experiments are not meaningful. The only tissue-specific RNAi experiments performed by the authors that use

generally accepted methodology are limited to those using the VP303 strain in the analysis of intestine-specific insulins, as noted in my initial review. It may be helpful for the authors to review the rationale for the use of these strains (and other strains utilized for tissue-specific RNAi). To be clear, the issue is not "off-target effects" stated by the authors in their response, but rather, the targeted gene will be affected in a tissue and cell type distinct from what was intended.

The authors refer to "multiple orthogonal lines of evidence," but the only ones that relate to neuron-specific issues may be the use of cell-specific rescue, which is spotty and only done in limited instances, and in any case, does not establish necessity, only supports sufficiency. The key results derived from what the authors perceived were neuron-specific RNAi experiments are not supported by "multiple orthogonal lines of evidence."

The authors refer to their prior 2017 study that utilized the same methodology. I am not aware of other manuscripts using the same method, though the concerns stated here would still apply. Given the experimental concerns outlined here, the authors' 2017 manuscript also invites considerable skepticism.

Response:

We thank the reviewer for continued engagement and for highlighting concerns regarding the interpretation of RNAi-based experiments. We agree that inter-tissue movement of dsRNA has been documented, including the work from Ravikumar et al. We appreciate the opportunity to clarify both the scope of our conclusions and the role of RNAi in this study. First, our primary conclusions regarding intestinal INS-37 and INS-7 signaling are based on experiments using the intestine-specific RNAi strain VP303, which is widely accepted for restricting RNAi efficacy to the intestine. These experiments form the foundation of our claims regarding intestinal ILP regulation. Second, conclusions related to neuronal mechanisms are not based on RNAi alone. Rather, RNAi-based perturbations are interpreted in conjunction with multiple independent approaches, including loss-of-function mutants, tissue-specific rescue experiments, CRISPR knock-in reporters, and direct functional measurements of neuronal activity and behavior. We agree that cell-specific rescue experiments establish sufficiency rather than necessity, and we have therefore framed these results as supportive rather than definitive on their own.

In response to the reviewer's concerns and following editorial guidance, we have now explicitly discussed the limitations of cell-restricted RNAi approaches and the potential contribution of low-level silencing in unintended tissues in the revised Discussion as follows: "A limitation of this study is the use of RNAi to infer cell-

specific gene function. Previous studies have shown that dsRNA can spread between tissues⁶⁵, and therefore low-level silencing in unintended cells cannot be excluded. Accordingly, RNAi-based results were interpreted in conjunction with complementary approaches, including genetic mutants, tissue-specific rescue, CRISPR knock-in reporters, and functional imaging and behavioral analyses.”

We hope that this clarification, together with the added Discussion text, addresses the reviewer’s concerns while accurately reflecting both the strengths and limitations of the experimental approaches used in this study.

Reviewer #2 (Remarks to the Author):

The authors fully addressed my questions in their revised manuscript.

Response: We are pleased that the revisions have fully addressed the reviewer’s questions.

Reviewer #3 (Remarks to the Author):

I think that the authors have adequately addressed the comments made by the reviewers in the revised version of the manuscript. Therefore, I have no further comments.

Response: We thank the reviewer for this positive assessment and are pleased that the revisions adequately address the comments raised.

Reviewer #4 (Remarks to the Author):

The authors have addressed my concerns for the most part. I remain a bit skeptical of the overall model for the reasons stated in my original review, but if the authors are convinced then perhaps it makes sense to publish and let the field digest the results.

Response: We appreciate the reviewer’s careful evaluation of the revised manuscript and their candid perspective. While we acknowledge that aspects of the proposed model will benefit from further investigation, we believe that the data presented provide a coherent and well-supported framework. We are grateful for the reviewer’s thoughtful consideration and are pleased that the revisions have addressed the major concerns raised.